# Distinct oxygenation modes of the Gulf of Oman during the past 43,000 years – a multi-proxy approach

Nicole Burdanowitz[1], Gerhard Schmiedl[1], Birgit Gaye[1], Philipp M. Munz[2], Hartmut Schulz[2]

[1] Institute for Geology, Center for Earth System Research and Sustainability (CEN), Universität Hamburg, Bundesstraße 55, 20146 Hamburg, Germany
[2] Department of Geosciences, Eberhard Karls Universität Tübingen, Hölderlinstr. 12, 72074 Tübingen, Germany

*Correspondence to*: Nicole Burdanowitz (nicole.burdanowitz@uni-hamburg.de)

**Abstract.**

Changing climatic conditions can shape the strength and extent of the oxygen minimum zone (OMZ). The presence and variability of the OMZ in the Arabian Sea is of importance for its ecosystem. The state of oxygenation has, for instance, an impact on the pelagic and benthic faunal community or the nitrogen and carbon cycles. It is important to understand the dynamics of the OMZ and related marine environmental conditions because of its climate feedbacks. In this study, we combined three independent proxies to reconstruct the oxygenation state of the water column and bottom water in the Gulf of Oman for the past about 43 ka. This multi-proxy approach is done for the first time at the northeast Oman margin located in the Gulf of Oman. We used bulk sedimentary nitrogen isotopes ($\delta^{15}N$) as well as the alkane ratio (lycopane + $n$-$C_{35}$)/$n$-$C_{31}$ and benthic foraminiferal faunal analysis to reconstruct the strength of the OMZ in the water column and bottom water oxygenation, respectively. Our results show that the Gulf of Oman experienced strong pronounced OMZ and bottom water deoxygenation during the Holocene. In contrast, during the Last Glacial Maximum (LGM)/ Marine Isotope Stage (MIS) 2 the Gulf of Oman was very well ventilated with a highly diverse benthic foraminiferal community. This may have been caused by stronger wind-induced mixing and better ventilation by oxygen-rich water masses. Our results also show moderate oxygenation during MIS 3 with deoxygenation events during most of the warmer Dansgaard-Oeschger (D/O) events. We propose two distinct oxygenation modes for the Gulf of Oman: 1) a stable period of either strongly pronounced water column OMZ and bottom water deoxygenation or well-oxygenated water column and bottom water conditions and 2) an unstable period of oscillating oxygenation states between moderately oxygenated (stadials) and deoxygenated (interstadial, D/O events) conditions. The unstable period may be triggered by an interstadial AMOC mode, which is required to initiate D/O events.

## 1 Introduction

Oxygen minimum zones (OMZs) are an important feature of the oceans. Although they cover only about 8 % of the total oceanic surface (Paulmier and Ruiz-Pino, 2009), they have a great impact on the marine environment and the carbon and nitrogen cycles (Friederich et al., 2008; Gruber, 2004; Paulmier et al., 2011; Rixen et al., 2020; Wakeham, 2020). For instance, denitrification plays a major role in the nitrogen cycle of OMZs and denitrification also acts as the largest sink for nitrogen in

the ocean (Gruber, 2004; Rixen et al., 2020). Moreover, OMZs have the potential to act as a source for greenhouse gases like $N_2O$, $CO_2$ or $CH_4$ (Friederich et al., 2008; Naqvi et al., 2010; Paulmier and Ruiz-Pino, 2009). Further, several studies show that the intensification and deoxygenation of OMZs are positively coupled to climate warming (e.g. Bopp et al., 2013; Breitburg et al., 2018; Busecke et al., 2022; Zhou et al., 2022). One of the strongest OMZ is located in the Arabian Sea

comprising about 17 % of the worldwide area of OMZs (Rixen et al., 2020). Over half of the permanently hypoxic shelf sediments are located in the northern Indian Ocean affecting benthic ecosystems, for instance, the diversity of benthic foraminifera (Helly and Levin, 2004; Levin, 2003). Although the diversity of benthic foraminifera is generally reduced under low oxygen conditions, the OMZ faunas of the Arabian Sea seem to be more resilient to low oxygen conditions when compared to other non-OMZ regions. This higher resilience can be attributed to a dominance of nitrate respiring infaunal benthic

foraminifera in OMZ regions (Schmiedl et al., 2023).

  The Gulf of Oman, as part of the Arabian Sea, is a complex and poorly understood region. It is influenced by the intrusion of warm, highly saline and more oxygenated Persian Gulf Water (PGW), when compared to the Arabian Sea water. Different gyres in the Gulf of Oman and their seasonal variations lead to a patchy distribution of the PGW in this area (Pous et al., 2004; Wang et al., 2013). Little is known about the oceanic environmental conditions in the past and its response to climate change

in this region as only a few sedimentary records and modelling studies exist (Cullen et al., 2000; Lachkar et al., 2019; Schmidt et al., 2020; Sirocko et al., 1991; Staubwasser et al., 2003). According to Lachkar et al. (2019), rising sea surface temperatures (SSTs) in the Persian Gulf, especially during the winter month, reduce the ability of the PGW to ventilate the upper OMZ in the Gulf of Oman due to shallower spreading of the PGW. The authors also concluded that a strengthening of the OMZ in the Gulf of Oman can be also induced by lower sea level. In this scenario, the Persian Gulf has fallen dry and, therefore, the

ventilation via the PGW would stop completely (Lachkar et al., 2019). However, existing Holocene and late Pleistocene sedimentary records (cores M5/3a – 422 & Orgon4-KS8, Sirocko et al., 1991, 2000; Staubwasser et al., 2003) located below the OMZ in the Gulf of Oman do not investigate the oxygenation status in this area. Nevertheless, from sedimentary records in other parts of the Arabian Sea OMZ we know that the OMZ varied in its extend and volume during the past (Altabet et al., 2002; Böll et al., 2014; Burdanowitz et al., 2019; Gaye et al., 2018; Ivanochko et al., 2005; Möbius et al., 2011; Pichevin et

al., 2007; Suthhof et al., 2001). Those records have shown a greater expansion of the OMZ during the Holocene (interstadial), also known as Marine Isotope Stage 1 (MIS 1), than during the Last Glacial Maximum (LGM, MIS 2) and stadials of MIS 3. Further, most of the rapid warming events of the Dansgaard-Oeschger (D/O) events during the last glacial period resulted in a strengthening of the OMZ, albeit each single D/O event had a different impact on local scale (Altabet et al., 2002; Ivanochko et al., 2005; Pichevin et al., 2007). However, due to the lack of continuous high-resolution archives in the Gulf of Oman, little

is known about the OMZ and bottom water oxygenation prior to the Holocene, especially without the influence of Persian Gulf Water (PGW) in this area.

  Most of the records are based on the qualitative reconstruction of the oxygenation status. The most common used proxy is the bulk $\delta^{15}N$ of marine sediments. The global average $\delta^{15}N$ of deep-water nitrate is about $4.8 \pm 0.2$ ‰ but in intermediate and surface water the $\delta^{15}N$ of nitrate can deviate strongly from this value due to the isotopic fractionation associated with nitrogen

cycling processes (Sigman et al., 2000; Sigman and Fripiat, 2019). In oxygen depleted environments heterotrophic bacteria use nitrate as an electron acceptor to oxidize organic matter and reduce the nitrogen in several steps to nitrogen gas (Sigman and Fripiat, 2019). Denitrification is associated with a strong isotopic effect of 15-25 ‰ so that the residual nitrate becomes strongly enriched in $^{15}$N and can thus have $\delta^{15}$N values >20 ‰ (Casciotti, 2016). Upward mixing and upwelling can transport this enriched isotopic signal where it is incorporated into phytoplankton. During the nitrate consumption, phytoplankton also

uses the lighter $^{14}$N with an isotope discrimination factor of about 5 ‰ and incorporate the denitrification signal into the biomass (Montoya, 2008). After decease of the biomass and sinking to the ocean floor, the denitrification signal is transported to the seafloor and can be stored in the sediments (e.g. Altabet et al., 1995; Gaye-Haake et al., 2005). Therefore, $\delta^{15}$N values can be used as a proxy for the OMZ strength in the water column (e.g. Altabet et al., 1995; Reichart et al., 1997). However, diagenetic processes can alter sedimentary $\delta^{15}$N values especially under low sedimentation rates (Gaye-Haake et al., 2005;

Jung et al., 1997; Junium et al., 2015; Möbius et al., 2011; Tesdal et al., 2013). Nevertheless, it was found that in the Arabian Sea, especially under high sedimentation rates on the shelf and slope, sedimentary $\delta^{15}$N values are a reliable indicator for past denitrification processes (Möbius et al., 2011).

Another rarely used biomarker is the isoprenoid hydrocarbon lycopane, which is mainly limited to sediments from OMZs or oceanic anoxic events in the past (van Bentum et al., 2009; Dummann et al., 2021; Farrington et al., 1988; Ogihara, 2014;

Sabino et al., 2020, 2021; Sinninghe Damsté et al., 2003). It is discussed that probably photoautotrophic organisms produce lycopane, but its origin is not yet fully resolved (Freeman et al., 1994; Sinninghe Damsté et al., 2003; Wakeham et al., 1993). However, these studies have shown that lycopane is well preserved in anoxic environments and is fast degraded under oxic conditions. Lycopane often co-elutes with the $n$-alkane $n$-$C_{35}$ (van Bentum et al., 2009; Naeher et al., 2012; Sabino et al., 2020; Sinninghe Damsté et al., 2003), with the latter mainly produced by terrestrial plants (e.g. Eglinton and Hamilton, 1967; Meyers

and Ishiwatari, 1993). An unusual high abundance of $n$-$C_{35}$ and lycopane (if co-eluted) or lycopane alone were detected in some parts of the Arabian Sea with highest contents within the OMZ (Schulte et al., 1999, 2000; Sinninghe Damsté et al., 2003). Therefore the (lycopane + $n$-$C_{35}$)/$n$-$C_{31}$ ratio can be used as indicator for bottom water oxygenation with ratios > 0.5 representing suboxic to anoxic conditions (van Bentum et al., 2009; Sinninghe Damsté et al., 2003). Whereas these two presented proxies are a qualitative way to reconstruct the OMZ strength and bottom water oxygenation, benthic foraminifera

can be used to achieve a quantitative oxygenation status reconstruction. The different benthic foraminifera species with their broad specific oxygen level requirements are widely used to reconstruct past bottom water oxygen conditions (e.g. Fontanier et al., 2002; Jorissen et al., 1995; Koho et al., 2008; Kranner et al., 2022; Lu et al., 2023; Schmiedl et al., 2000, 2010).

Here, we present the first high-resolution record reconstructing the OMZ strength and oxygen status of the bottom water at a depth situated within the present OMZ for the Holocene and late Pleistocene (past approx. 43 ka) at the Oman margin in the

Gulf of Oman by using a multi-proxy approach.

## 2 Modern oceanographic setting in the study area

The Gulf of Oman as part of the northwestern Arabian Sea connects the Arabian Sea via the Strait of Hormuz with the Persian Gulf. The northern Arabian Sea and the Gulf of Oman are affected by different water masses. At the surface, the Arabian Sea High-Salinity Water (ASHSW) with salinities between 35.3 and 36.7 comprises the upper 200 m (Kumar and Prasad, 1999; Shenoi et al., 1993). The highly saline (> 36.5) low-oxygen PGW is responsible for a weak ventilation of the upper OMZ (Figure 1, 150-300 m water depth), especially in the southern part of the Gulf of Oman (Bower et al., 2000; Kumar and Prasad, 1999; Morrison et al., 1998; Pous et al., 2004; Prasad et al., 2001; Schmidt et al., 2020; Shenoi et al., 1993; Wang et al., 2013). The Red Sea Water (RSW, salinity: 35.1 – 35.6) enters the Arabian Sea via the Gulf of Aden and is found at intermediate water depths between 600 to 900 m (Bower et al., 2000; Kumar and Prasad, 1999). The RSW is transported northward along the Oman coast, especially during the SW monsoon (Acharya and Panigrahi, 2016; Beal et al., 2000; Kumar and Prasad, 1999; Schmidt et al., 2020). Other intermediate water masses (500 – 1000 m water depth) are the Indian Central Water (ICW), which is a mixture of the Antarctic Intermediate Water (AAIW) and the Indonesian Intermediate Water (IIW), and enters the Arabian Sea during the SW monsoon via the Somali current along the Somali and Arabian coasts (Acharya and Panigrahi, 2016; Schott and McCreary, 2001). The initial oxygen-rich ICW becomes less oxygenated during its transport to the Arabian Sea (Acharya and Panigrahi, 2016). However, the oxygen content is still higher than those of the PGW and RSW (Rixen et al., 2014).

The region is influenced by different atmospheric circulations, which result in different seasonal primary productivity patterns (Figure 2). During the summer season, strong southwest (SW) winds and the Findlater Jet in the western Arabian Sea induce upwelling and lateral transport of cold nutrient-rich water resulting in high primary productivity (Figure 2b) (Andruleit et al., 2003; Haake et al., 1993; Rixen et al., 2020). In contrast, highest productivity in the northern Arabian Sea and Gulf of Oman is observed during the winter season (Figure 2d) when high amounts of nutrients are available due to deeper wind-induced surface mixing by the NE winds (Böll et al., 2014; Madhupratap et al., 1996; Rixen et al., 2020). During the summer season, the Ras Al Hadd Jet at the northeastern corner of Oman can transport cold upwelled water into the Gulf of Oman via eddies (Schott and McCreary, 2001). Further, eddies generated in the Gulf of Oman transport PGW trapped in its core into the Arabian Sea (L'Hégaret et al., 2015; de Marez et al., 2019). Both may have an impact on the sea surface temperature, salinity and primary productivity in this region.

## 3 Material and methods

The 739 cm long gravity core SL167 was collected in the northwestern Arabian Sea from the northeastern Oman margin off northern Oman (Gulf of Oman, 22°37.15'N, 59°41.49'E, 774 m water depth) during RV *Meteor* cruise M74/1b in 2007 (Figure 1). In total, 371 samples were used for $\delta^{15}N$ analyses with continuous sampling intervals of 2 cm. For lipid biomarker analyses 225 samples were used with an interval of 2 cm in the upper 162 cm of the core and, due to low total organic carbon content (unpublished), an interval of 4 cm below 162 cm. All sediment samples were freeze-dried and homogenized with an agate

mortar and pestle prior to chemical treatment for the analyses. For the analyses of benthic foraminifera, 149 samples taken every 5 cm were used.

## 3.1 Dating & age model

The age model is based on twenty-one radiocarbon datings by accelerator mass spectrometry (AMS) measurements of surface-dwelling planktonic foraminifera, comprising *Globigerinoides spp.*, *Globigerinella spp.* and *Orbulina universa*, at Beta Analytic Inc, Miami, USA. The age-depth model was developed by using the Bayesian model package BACON v.2.5.6 (Blaauw and Christen, 2011) within R statistical software (v.4.3, R Core Team (2023)). We used the default settings except for accumulation mean (set to 50), memory mean (set to 0.3) and the calibration curve (set to Marine20). We applied a deltaR

($\Delta R$) of $93 \pm 61$ years. The $\Delta R$ is based on the weighted mean of two regional marine reservoir corrections (Muscat) by Southon et al. (2002) using the marine calibration database (Reimer and Reimer, 2001, http://calib.org/marine/).

## 3.2 Lipid biomarker analyses

Total lipid extracts (TLE) of about 3 to 18 g of the sediment samples were obtained by using a DIONEX Accelerated Solvent Extractor (ASE 200) at 100°C and 1000 PSI for 5 minutes (3 cycles) using a solvent mixture of dichloromethane:methanol

(DCM:MeOH, 9:1) following the procedure described in Herrmann et al. (2016). Prior to extraction a known amount of an internal standard (squalene) was added to the samples. After the extraction, TLEs were concentrated via rotary evaporation and transferred to combusted 4 ml vials. We have used $NaSO_4$ column chromatography to separate the TLEs into a hexane-soluble and hexane-insoluble fraction. Therefore, a combusted Pasteur pipette was packed with cleaned cotton wool and about 2 cm $NaSO_4$. The column was then first cleaned with about 8 ml hexane. Then TLE was transferred with about 1 ml hexane

to the column and the remaining 4 ml vial was cleaned 3 times with about 1 ml hexane, which was also transferred to the column. For the hexane-insoluble fraction, about 1 ml of DCM was added for 4 times on the column. The hexane-soluble was saponified (85°C, 2 h) in a 5 % potassium hydroxide (KOH) in MeOH solution and the neutral fraction was extracted with hexane. To obtain the *n*-alkane containing apolar fraction from the neutral fraction, a column chromatography packed with deactivated silica gel (5 % $H_2O$, 60 mesh) was carried out by using hexane as solvent. Afterwards, the apolar fraction was

cleaned with hexane via column chromatography packed with silver nitrate coated silica gel ($AgNO_3$-Si).
For quantification of the *n*-alkanes a Thermo Scientific Trace 1310 gas chromatography coupled to a flame ionization detector (GC-FID) equipped with a Thermo Scientific TG-5MS column (30 m, 0.25 mm, 0.25 µm). The carrier gas $H_2$ with a flow rate of 35 ml min$^{-1}$ was used. The PTV injector was operated starting at 50°C ramped with 10°C s$^{-1}$ to 325°C in a splitless mode. The initial GC temperature was programmed to 50°C (held 3 min) and then to increase with 6°C min$^{-1}$ to a final temperature

of 325°C, which was held for 20 minutes. For quantification of *n*-alkanes an external standard containing the $n\text{-}C_8 - n\text{-}C_{40}$ alkanes was used in a known concentration. Quantification precision of repeated analyses of the external standard was 8 %.
The mass spectra of two samples (14 – 16 cm and 182 – 186 cm) were investigated with a Thermo Scientific Trace GC Ultra coupled to a Thermo Scientific DSQ II mass spectrometer (GC-MS) with He (2 ml min$^{-1}$ flow rate) as carrier gas. The initial

GC temperature of 50°C was held for 3 minutes and ramped with 6°C min$^{-1}$ to 325°C with the final temperature hold for 25 minutes. To identify the compounds of the apolar fraction the mass spectra were compared with published mass spectral data. The average chain length (ACL) of plant-wax derived $n$-alkanes are commonly used to identify plant functional types and environmental conditions (e.g. Collister et al., 1994; Cooper et al., 2015; Eglinton and Eglinton, 2008; Rommerskirchen et al., 2006). For instance, plants located in arid environments tend to have longer ACL than plants in humid environments (e.g. Carr et al., 2014; Rommerskirchen et al., 2006; Vogts et al., 2009). But the ACL also differs with the plant functional type, for instance C4 grasses have in general higher ACL than woody gymnosperms (e.g. Bush and McInerney, 2013; Carr et al., 2014; Cooper et al., 2015). However, the validity of the ACL is limited and, if possible, should be combined with compound-specific isotope measurements of the $n$-alkanes (e.g. Eglinton and Eglinton, 2008; Rommerskirchen et al., 2006; Vogts et al., 2009). We have calculated the ACL of $n$-alkanes using following equation:

$$ACL_{27-33} = \frac{27 \times C_{27} + 29 \times C_{29} + 31 \times C_{31} + 33 \times C_{33}}{C_{27} + C_{29} + C_{31} + C_{33}} \qquad (1)$$

The carbon preference index (CPI), an indicator for the odd-over-even predominance, is usable to distinguish between terrestrial plant and petroleum sources (e.g. Bray and Evans, 1961; Cranwell, 1978, 1981; Pancost and Boot, 2004). Terrestrial plant derived $n$-alkanes and recent sediments with unaltered organic material have an odd-over-even predominance with a CPI higher than 3 (e.g. Bray and Evans, 1961). In contrast, petroleum sources have higher abundance of even $n$-alkanes with CPIs < 1 and are an indication of higher degradation (e.g. Bray and Evans, 1961). The CPI was calculated by using the following equation:

$$CPI_{27-33} = 0.5 \times \left( \frac{C_{27} + C_{29} + C_{31} + C_{33}}{C_{26} + C_{28} + C_{30} + C_{32}} + \frac{C_{27} + C_{29} + C_{31} + C_{33}}{C_{28} + C_{30} + C_{32} + C_{34}} \right) \qquad (2)$$

where Cx is the concentration of the $n$-alkane with x atoms.

### 3.3 Bulk nitrogen isotope ($\delta^{15}$N) analyses

For the $\delta^{15}$N analyses, following the procedure described in Menzel et al., (2014), 6 to 61 mg of the freeze-dried and homogenized sediment were weighted into Sn-capsules. $\delta^{15}$N were obtained by combusting the samples at 1050°C in a Thermo Scientific Flash EA1112 elemental analyser coupled to a Finnigan MAT 252 isotope ratio mass spectrometer. Nitrogen was calibrated against the International Atomic Energy Agency (IAEA) reference standard IAEA-1 and IAEA-2, respectively. In addition to an internal standard, both (IAEA-1 and IAEA-2) were used as working standards. Replicate measurements of these standards yielded a precision better than 0.2 ‰. The samples were measured in duplicate with a mean standard deviation of 0.07 ‰.

### 3.4 Benthic foraminifera analyses

For faunal analyses, between 198 and 558 (average of 311) individuals were at each depth counted from representative splits of the size fraction >125 µm. Species assignment is mainly based on Jones (1994), Den Dulk (2000), Szarek (2001),

Schumacher et al. (2007) and Debenay (2012). Allochthonous shelf taxa, including larger benthic foraminifera (e.g., genera *Amphistegina*, *Heterostegina*, *Borelis*, *Peneroplis*), typical neritic taxa (e.g., *Ammonia* spp., *Cibicides refulgens*, *Elphidium* spp., *Lobatula lobatula*), and taxa with floating chambers (genera *Cymbaloporetta*, *Millettiana*, *Tretomphalus*, *Tretomphaloides*) were removed from the census data set (Murray, 1991). For benthic foraminiferal diversity, the Shannon Index H(S) was calculated according to Buzas and Gibson (1969). The H(S) considers the number of species and their relative proportion in the sample. The H(S) value is at a maximum, when all species have equal proportions, while species with low abundances contribute little to it. In eutrophic to mesotrophic ecosystems, the diversity and microhabitat structure is oxygen-controlled, as predicted by the TROX (Trophic-Oxygen) model (Gooday, 2003; Jorissen et al., 1995) (Figure 3). For the estimation of bottom-water oxygenation, the different taxa were classified into oxic, suboxic, and dysoxic taxa based on their modern microhabitat preferences (Table S1) (Schmiedl et al., 2023). Bottom-water oxygen concentrations were then calculated based on the Enhanced Benthic Foraminiferal Oxygen Index and associated transfer function of Kranner et al. (2022) with the modification of Schmiedl et al. (2023). The mean standard deviation (SD) across the entire oxygen range (0-6 ml $l^{-1}$) is ±0.61 ml $l^{-1}$. However, SD is lower at suboxic to dysoxic conditions, with SD of ±0.49 and ±0.08 ml $l^{-1}$ across oxygen ranges of 1-2 ml $l^{-1}$ and 0-1 ml $l^{-1}$, respectively. The expected relation between oxygen concentration and benthic foraminiferal diversity is clearly expressed at core site SL167, since it is influenced by comparatively high organic matter fluxes and low oxygen conditions (Figure 3).

**3.5 Statistics**

To identify periodicities in our data sets we performed spectral and wavelet analyses in R (v.4.3, R Core Team (2023)). For the spectral analysis, we used the REDFIT function of the package dplR v.1.7.4 (Bunn et al., 2022; Bunn, 2008, 2010) which is based on the Fortran 90 REDFIT source code by Schulz and Mudelsee (2002). For the wavelet analysis the data sets were first interpolated to an even spaced data set by using the package ncdf4.helpers v.0.3-6 (Bronough, 2021) and the approx. function. The wavelet analysis were performed with the package biwavelet v.0.20.21 (Gouhier et al., 2021) using the morlet wavelet function and bias-corrected power spectrum which is based on Torrence and Compo (1998).

**4 Results**

**4.1 Age model of SL167**

The results of the AMS radiocarbon measurements are shown in Table 1. The core SL167 comprises the last about 42.6 ka (Figure 4). However, the last 3 ka BP are missing in the core as the core top represents ca. 3.1 ka BP. The mean sedimentation rates range between 0.10 mm/year and 0.49 mm/year with higher sedimentation rates during the Holocene compared to the Pleistocene.

## 4.2 Lipid biomarker and $\delta^{15}N$ reconstructions

The samples of SL167 show a strong odd-over even carbon number predominance with CPI values ranging between 2 and 13.8 and show a dominance of land-plant derived long chain $n$-alkanes, where the $ACL_{27-33}$ varies between 30 and 31.2 (Fig. A1). In some samples, for land-plant derived $n$-alkanes, unusual high contents of the $n$-alkane $C_{35}$ were detected in the GC-FID chromatograms. We chose one exemplary sample with a very high content of $n$-$C_{35}$ and one sample with a usual distribution of land plant derived $n$-alkanes for mass spectral analyses, respectively. For the latter one we found only the mass spectra of the $n$-alkane $C_{35}$ at the expected retention time. However, for the sample with the high $n$-$C_{35}$ content and unusual $n$-alkane distribution pattern, we found both, the mass spectra of lycopane (2,6,10,14,19,23,27,31-octamethyldotriacontane) and of $n$-$C_{35}$ (Figure 5). The characteristic fragments of lycopane are the ions m/z 113, 183, 253, 309, 337, 407 and 447 (Sinninghe Damsté et al., 2003), which are also detected in our sample with the co-elution of $n$-$C_{35}$ and lycopane (Figure 5b). It is known from other studies that $n$-$C_{35}$ and lycopane can co-elute during gas chromatography measurements (van Bentum et al., 2009; Sabino et al., 2020; Sinninghe Damsté et al., 2003).

The (lycopane + $n$-$C_{35}$)/$n$-$C_{31}$ varies between 0.1 and 1.7 throughout the core (Figure 6d). Highest ratios occur during the late Holocene (3 – 4 ka BP) and warmer Pleistocene periods reconstructed by the NGRIP ice core $\delta^{18}O$ (North Greenland Ice Core Project members, 2004) and Sofular cave $\delta^{13}C$ (Fleitmann et al., 2009) records, respectively.

The $\delta^{15}N$ values in our Gulf of Oman record vary between 4.7 and 9.2 ‰ (Figure 6b) with constant high values throughout the Holocene (> 7.2 ‰). The variability of $\delta^{15}N$ is high (4.5 ‰) during the Pleistocene with higher values > 7 ‰ during warmer phases.

## 4.3 Benthic foraminiferal diversity and reconstructed bottom-water oxygenation

Overall, the benthic foraminiferal diversity is high and ranges between H(S)=1.94 and 4.27 (average of 3.31 ± 0.64 SD). Significant fluctuations occur in the older part of the core with minima during interstadials, while Heinrich events H2 and H1, and the Last Glacial Maximum are characterized by comparatively high H(S) values (Figure 6). During the Holocene, the H(S) values exhibit a gradual decrease. The reconstructed oxygen values range between 0.27 and 3.25 (average of 1.46 ± 0.74 SD) and are negatively correlated to the benthic foraminiferal diversity and $\delta^{15}N$. D/O events, the B/A and Holocene are characterized by relatively low oxygen concentrations while cold intervals, particularly the interval around H1, exhibit higher values (Figure 6).

## 5 Discussion

The $\delta^{15}N$ is often used as indicator for past denitrification processes and, in the Arabian Sea, strength of the OMZ (e.g. Altabet et al., 2002; Gaye et al., 2018; Ivanochko et al., 2005; Möbius et al., 2011; Pichevin et al., 2007). However, the $\delta^{15}N$ signal can be biased by different processes, which need to be considered. One factor is a diagenetic overprint of $\delta^{15}N$ leading to an

enrichment of the heavier $^{15}N$ isotope, especially under low sedimentation rates (3 – 4 cm kyr$^{-1}$) (Altabet et al., 1999; Gaye-
Haake et al., 2005; Jung et al., 1997; Junium et al., 2015; Möbius et al., 2011; Tesdal et al., 2013). In contrast, shelf and slope
sediments deposited under high sedimentation rates in the Arabian Sea are considered to have unaltered $\delta^{15}N$ values and,
therefore, a reliable indicator for past denitrification processes (Möbius et al., 2011). The sedimentation rates are high in
SL167and varies between 9.7 and 49 cm kyr$^{-1}$. Further, they correlate with the nitrogen content but not with $\delta^{15}N$ values (Fig.
A2). The correlation of the nitrogen content and $\delta^{15}N$ may due to higher productivity and/or better preservation of organic
matter under sub-/anoxic conditions. Therefore, we argue that sedimentary $\delta^{15}N$ in SL167 are a reliable indicator for past
denitrification process. However, it may be possible that the $\delta^{15}N$ signal in SL 167 is affected by changes in the water mass
transport, mixing or lateral transport from e.g. the Oman Upwelling area. 5.1 Strength of the OMZ and bottom water
oxygenation in the Gulf of Oman

### 5.1.1 Pleistocene

Both independent proxies for bottom water oxygenation, (lycopane + $n$-C$_{35}$)/$n$-C$_{31}$ ratio, the O$_2$ reconstruction by benthic
foraminifera, and the Shannon diversity index, are in very good agreement at least for the Pleistocene part of the core (Figure
6a, c & d). Phases of sub-/anoxic bottom water match well with observed interstadials, the D/O events, in the Greenland ice
core records, e.g. NGRIP (North Greenland Ice Core Project members, 2004), except for D/O 6 and 7. Highest reconstructed
bottom water oxygen content (up to 3.2 ml l$^{-1}$) occurred from about 14.3 to 19.6 ka BP and during the Younger Dryas between
11.2 and 12.3 ka BP, respectively (Figure 6c). Further, MIS 3 (about 25 ka – 43 ka BP), is characterized by several distinct
peaks of a strong OMZ (Figure 6b) in the water column. Note, that we used the Sofular $\delta^{13}C$ cave record from Turkey
(Fleitmann et al., 2009) instead of the NGRIP to allocate the distinct D/O events as it seems to be more appropriate (Figure 7).
Fleitmann et al. (2009) found a systematic age offset for most D/O events between the NGRIP and the more regional Sofular
and Hulu cave (China) records by several centennial years, with younger ages for the D/O events in the NGRIP record. The
D/O events 3-5 and 8-11 are well pronounced as strong abrupt OMZ intervals in the Gulf of Oman. In contrast, in the nearby
Oman upwelling (core RC27-14) all D/O events, except 11, are expressed as strong OMZ intervals in the water column (Figure
7c) (Altabet et al., 2002). A high-resolution record (MD04-2876) in the northern Arabian Sea shows a similar pattern as core
RC27-14 (Pichevin et al., 2007), whereas a record (NIOP905) from the more southern Somali Upwelling area (Ivanochko et
al., 2005) does not show a strong OMZ in the water column during the first four D/O events (Figure 7b & d). Note that shifts
in OMZ peaks of the different records in the Arabian Sea may be due to age uncertainty as well as peak tuning of the other
records with the Greenland ice core (GISP2) $\delta^{18}O$ record.

Bottom water oxygenation may not only be driven by processes in the upper water column but also by the occurrence of
different water masses. The RSW is transported northward especially during the SW monsoon (Acharya and Panigrahi, 2016;
Beal et al., 2000; Kumar and Prasad, 1999) and may undergo oxygen depletion due to higher organic matter supply and its
decay on its way to the Gulf of Oman (Pathak et al., 2021). Acharya and Panigrahi (2016) have shown, that today's RSW core
extends between 600 and 660 m reaching nearly the seafloor at the core site (~740 m). Thus, phases with stronger SW

monsoons, such as the D/O interstadials, are characterized by more input of low oxygen RSW and, therefore, sub-/anoxic conditions of bottom water in the Gulf of Oman. This corroborates findings from the Oman Upwelling area, where oxygen bottom water reconstructions using benthic foraminifera show oxygen depleted conditions during phases with higher outflow of RSW into the northwestern Arabian Sea (Pathak et al., 2021).

The missing D/O 6 and weak D/O 7 events might be a result of the combination of local/regional and global conditions. First, the strength of the Indian and Asian Monsoon systems are strongly coupled to high latitude climate (e.g. Deplazes et al., 2013, 2014; Overpeck et al., 1996; Schulz et al., 1998; Singh et al., 2011; Wang et al., 2001). Phases with strong monsoon intervals are coupled to the D/O events found in Greenland ice cores. However, duration and strength of both varied through the time. In some records from the Asian region, D/O 6 had a weaker environmental impact than other D/O events (e.g. Deplazes et al., 2014; Ivanochko et al., 2005; Wang et al., 2001). Second, the fluctuation of the Red Sea sea level (Figure 7f) could lead to variations in the exchange of the RSW and the Indian Ocean/Arabian Sea, with stronger influence during higher sea level and vice versa (Arz et al., 2007; Siddall et al., 2003). However, the sea level was in general lower during MIS 3 than during the Holocene, which might led to a weaker influence of the RSW compared to recent conditions (Rohling et al., 2008; Siddall et al., 2003). Third, the degree of northward extension of the oxygen-rich AAIW is linked to North Atlantic climate (Jung et al., 2009; Pahnke and Zahn, 2005). The maxima in the northward extension and, therefore, ventilation of the Arabian Sea water column are found during the Heinrich stadials. In contrast, minima extension occurred during interstadials with a stronger monsoon (Jung et al., 2009). In total, the interplay of this three mentioned factors, the bipolar seesaw structure of the northern and southern hemispheric climate but also the climate and oceanic patterns on regional scale (e.g. Lemieux-Dudon et al., 2010; Stocker and Johnsen, 2003), may lead to the feature that some of the D/O events as well as Heinrich events (H4) are not represented in the Gulf of Oman record. Further, stronger northwesterly as well as northeasterly winds during stadials (Leuschner and Sirocko, 2000) could have induced stronger deep winter mixing compared to present and, therefore, better ventilation (Lu et al., 2023; Reichart et al., 1998).

A striking feature of our Gulf of Oman $\delta^{15}N$ record is the prominent triple peak of strong water column deoxygenation from D/O events 3 to 5 (Figure 6b). This is in contrast to other Arabian Sea records, not showing a prominent triple peak from D/O events 3 to 5 but 5 to 7 with a broader D/O 8 event beforehand (Schulz et al., 1998). Interestingly, at the core site sub-/anoxic bottom water conditions occurs during D/O 2, whereas the OMZ in the water column was not developed. We speculate that the inflow of oxygen depleted RSW into the Gulf of Oman was still strong and/or the maxima northward extension of the ventilating AAIW/ICW was further south, while denitrification processes were not as strong as during other pronounced D/O events. This is also the case for a record from the southern Somali Upwelling area (Figure 7d). However, along the Oman Upwelling area (Altabet et al., 2002), a strong SW monsoon could have favoured upwelling-induced denitrification and therefore a strengthening of the OMZ only in this region, albeit not as strong as during former D/O events.

In contrast to other parts of the Arabian Sea, the water column in the Gulf of Oman was well ventilated during the whole LGM indicated by low $\delta^{15}N$ values (Figure 7) and by high benthic foraminiferal diversity (Figure 6). The northern and northeastern Arabian Sea experienced a stronger and/or longer NE monsoon season during the LGM (Pichevin et al., 2007; Suthhof et al.,

2001). Thus, the stronger wind-induced mixing may have resulted in a better-ventilated OMZ compared to the interstadials. A recent study from the Oman Upwelling area also suggests slightly increased oxygenation in the upper water column (~600 – 820 m) during the LGM compared to the modern interstadial (Lu et al., 2023). Further, less lateral transport of nutrient- and oxygen-poor waters from the Oman Upwelling due to weakened SW winds was supposed to mitigate denitrification in the northern and northeastern Arabian Sea (Suthhof et al., 2001). In addition, a northward expansion of the well oxygenated AAIW during stadials results in a better-ventilated intermediate waters in the Arabian Sea, reducing the strength of the OMZ (Jung et al., 2009; Pichevin et al., 2007). However, the productivity was still high enough to maintain an OMZ. The absence of the OMZ at the core location from the LGM to the beginning of the Bølling/Allerød (B/A) may be due to more intense northwesterly winds in the Gulf of Oman and NE monsoon (Leuschner and Sirocko, 2000; Sirocko et al., 2000) leading to stronger wind-induced mixing and ventilation of the water column. Weak Indian summer monsoon phases occurred also during the LGM in NE India (Dutt et al., 2015) and the eastern Arabian Sea (Saravanan et al., 2020). In addition, Red Sea sea level was too low during that time (Sergiou et al., 2022; Siddall et al., 2003) to induce significant RSW influence on bottom water deoxygenation in the Gulf of Oman. The Persian Gulf plays no role in contributing water masses to the Gulf of Oman as it was nearly completely dry until the LGM and had no connection to the Gulf of Oman (Lambeck, 1996; Stoffers and Ross, 1979).

During the B/A interstadial, a strong OMZ and bottom water oxygenation occurred in the Gulf of Oman and was also observed for the whole Arabian Sea (Altabet et al., 2002; Ivanochko et al., 2005; Kessarkar et al., 2013; Orsi et al., 2017; Pichevin et al., 2007; Suthhof et al., 2001). Contrary to the $O_2$ reconstruction of the benthic foraminifera we see no distinct high ratios of (lycopane + $n$-$C_{35}$)/$n$-$C_{31}$ during the B/A. Here it is possible that the threshold of oxygen depletion was not reached to preserve lycopane in the sediment.

### 5.1.2 Holocene

The observed low reconstructed $O_2$ contents and high $\delta^{15}N$ values with minor fluctuations during the Holocene indicate a continuously strong OMZ and sub-/anoxic bottom water at the Oman margin area in the Gulf of Oman. This is in line with other studies for most parts of the Arabian Sea (Figure 7, Altabet et al., 2002; Burdanowitz et al., 2019; Gaye et al., 2018; Kessarkar et al., 2013; Pichevin et al., 2007; Suthhof et al., 2001). In the course of the post LGM sea level rise, the Persian Gulf became connected to the Gulf of Oman via the Strait of Hormuz at around 13 ka BP with flooding of the central part of the Persian Gulf until about 11.5 ka BP (Lambeck, 1996). At least, since the Mid-Holocene, the PGW is an important factor in ventilating the upper OMZ (150 – 300 m) in the northern Arabian Sea (Lachkar et al., 2019). However, modern data show a relatively low increase of $O_2$ content (about 0.5 mol l$^{-1}$) due to PGW at (Figure 1) and near the core location (Munz et al., 2017).

Highest ratios of (lycopane + n-$C_{35}$)/n-$C_{31}$ (> 1.0) occurred during the late Holocene, which may be due to relatively high total organic carbon mass accumulation rates during that time (Fig. A1) and thus is an effect of fast burial. This assumption is also corroborated by the increasing dominance of the opportunistic food indicator *Uvigerina peregrina* (Koho et al., 2008; Schmiedl et al., 2010) which favours high supply of organic matter, leading to decreasing benthic foraminiferal diversity.

Overall, our multi-proxy oxygen record for the Gulf of Oman shows striking features. First, during most interstadial periods (D/O events and Holocene) the denitrification/OMZ in the water column as well as bottom water deoxygenation was strong. Second, during MIS 3 oxygen conditions oscillate between moderately oxygenated (glacial) and deoxygenated/denitrification conditions (interglacial, D/O events) marking this period as an environmentally unstable period. Here, the oxygen concentrations were close to the threshold of suboxic/anoxic conditions and therefore fluctuated between oxia and suboxia driven by changes in monsoon strength. In contrast, LGM/MIS 2 and the Holocene period show quite constant oxygen and deoxygenation/denitrification conditions, respectively. We argue that these two periods are stable enough to suppress strong oscillations of oxygenation versus deoxygenation.

## 5.2 Periodic changes and potential global drivers for OMZ strength and bottom water oxygenation

In order to identify specific periodicities, we have performed spectral- and wavelet analyses for the $\delta^{15}N$ record (Figure 8) and the oxygen ($O_2$) reconstruction of the benthic foraminifera (Figure 9), respectively. For the strength of the OMZ in the water column based on $\delta^{15}N$, we have found significant (95 % confidence interval (CI)) periods of 3.8 ka, 3.2 ka, 2.7 ka, 1.3 ka, 1.1 ka as well as 690 and 570 years (Figure 8a). For the reconstructed $O_2$ bottom water variations, we have found significant periods (95 % CI) of 1.1 ka, 710, 670, 600 and 570 years as well as 3.2 ka (90 % CI, Figure 9a). These periods are mainly present between about 25 ka – 43 ka BP (MIS 3) and 8 ka – 16 ka BP (Transition LGM/MIS 2 to Holocene, Figure 8b & Figure 9b) for both proxies. We attribute the periods of about 1.1 – 1.3 ka as well as around 3.2 ka to the D/O events as they occur with periods of about 1.5 ka and 3.0 ka (Kuniyoshi et al., 2022; Schulz, 2002). Especially the period of around 1.5 ka is a widespread feature in climate records (e.g. Bond et al., 2001; Jaglan et al., 2021; Lauterbach et al., 2014; Leuschner and Sirocko, 2000; Saravanan et al., 2020; Thamban et al., 2007; Wang et al., 2005). However, Obrochta et al. (2012) question the general idea of a 1.5 ka oscillation as they propose a superposition of the 1.0 ka and 2.0 ka cycles, which is also evident in our record with the 1.1 to 1.3 ka periods. Another strong evidence for D/O events as trigger of these periods is that the D/O events are also absent during the LGM/MIS 2 period (Buizert and Schmittner, 2015; North Greenland Ice Core Project members, 2004). The cold phase and its high global ice volume during the LGM inhibited an interstadial AMOC mode (strong AMOC/warm North Atlantic), which is required to initiate D/O events (Buizert and Schmittner, 2015). An interesting feature is the successive decrease in duration and magnitude of the D/O events between two Heinrich events (Buizert and Schmittner, 2015). However, this pattern may or may not be the reason for the missing D/O 6 signal in our record as D/O 5 is strong and the last event between H4 and H3.

The periods of 570 to 710 years can be related to solar cycles (Liu et al., 2012; Stuiver and Braziunas, 1993; Wang et al., 2005a). These cycles are prominent worldwide and also found in the East Asian (Liu et al., 2012; Wang et al., 1999, 2005a; Xu et al., 2014) and Indian Monsoon (Burdanowitz et al., 2021; Neff et al., 2001; von Rad et al., 1999; Saravanan et al., 2020; Sarkar et al., 2000; Thamban et al., 2007) realms, respectively. Some authors attribute a 725-775 years periodicity to changes in the Indian summer monsoon (Saravanan et al., 2020), ENSO-like cycles (Russell et al., 2003) or a harmonic cycle of the

1500 year cycle (von Rad et al., 1999), linked to the thermohaline circulation (Wang et al., 1999) or to the ITCZ movement in the tropics (Russell and Johnson, 2005). Most of the high frequency periods (< 1000 years) are reported for the Holocene (Burdanowitz et al., 2021; Liu et al., 2012; Neff et al., 2001; von Rad et al., 1999; Russell et al., 2003; Russell and Johnson, 2005; Stuiver and Braziunas, 1993; Thamban et al., 2007; Wang et al., 1999, 2005b; Xu et al., 2014) or the LGM (Saravanan et al., 2020). Here we show that these high frequency periods are also present during MIS 3. The intensity and position of the monsoon system is driven by the summer insolation (precession signal) and the latitudinal temperature gradient ( obliquity signal), respectively, with the latter influenced by the latitudinal insolation gradient and ice cover (Davis and Brewer, 2009). The authors postulate that a strong latitudinal temperature gradient moves the monsoon system towards the equator, which was the case during the LGM/MIS 2 for the summer (ca. 20 – 28 ka BP) and winter (ca. 18 – 25 ka BP) latitudinal temperature gradients (Davis and Brewer, 2009). This coincides with the well-ventilated water column and bottom water in the Gulf of Oman during that time suggesting a response to the obliquity signal.

## 6 Conclusion

Our study presents a multiproxy approach reconstructing the water column and bottom water oxygenation in the Gulf of Oman for the past about 43 ka for the first time. The three independent proxies based on bulk sediment ($\delta^{15}$N), lipid biomarker analysis (ratio of (lycopane + $n$-C$_{35}$)/$n$-C$_{31}$) and benthic foraminifera taxa (Enhanced Benthic Foraminiferal Oxygen Index including transfer function of Kranner et al. (2022)) show a robust and mostly consistent pattern in our record. The Holocene is characterized by strong OMZ conditions and bottom water deoxygenation, which is consistent with other studies from the Arabian Sea. In contrast to other regions in the Arabian Sea, the water column and the bottom water in the Gulf of Oman were very well ventilated during the LGM/MIS 2. The well oxygenated conditions of the Gulf of Oman lead to a highly diverse benthic foraminiferal assemblage. We attribute the good ventilation to stronger wind-induced mixing of the water column and better ventilation by oxygen-richer oceanic currents leading to stronger northward intrusion of the AAIW, corroborating other recent studies. The OMZ strength and bottom water oxygenation reconstruction during MIS 3 reveal oscillating conditions of moderately-oxygenated (stadials) and deoxygenated conditions (interstadials, D/O events). Besides the prominent D/O cycles, we also found prominent high frequency periods (570 to 710 years) during MIS 3 by using spectral and wavelet analyses.

In consideration of our results, we propose two different modes of the oxygenation status for the past about 43 ka. The first mode is a stable period of either strong OMZ and bottom water deoxygenation (Holocene) or well oxygenation of the water column and bottom water (LGM/MIS 2). The second mode is an unstable period with oscillations between moderately oxygenated and deoxygenated conditions, respectively. This is visible in the MIS 3 period of our record with fluctuating high and low oxygen conditions.

## Appendix

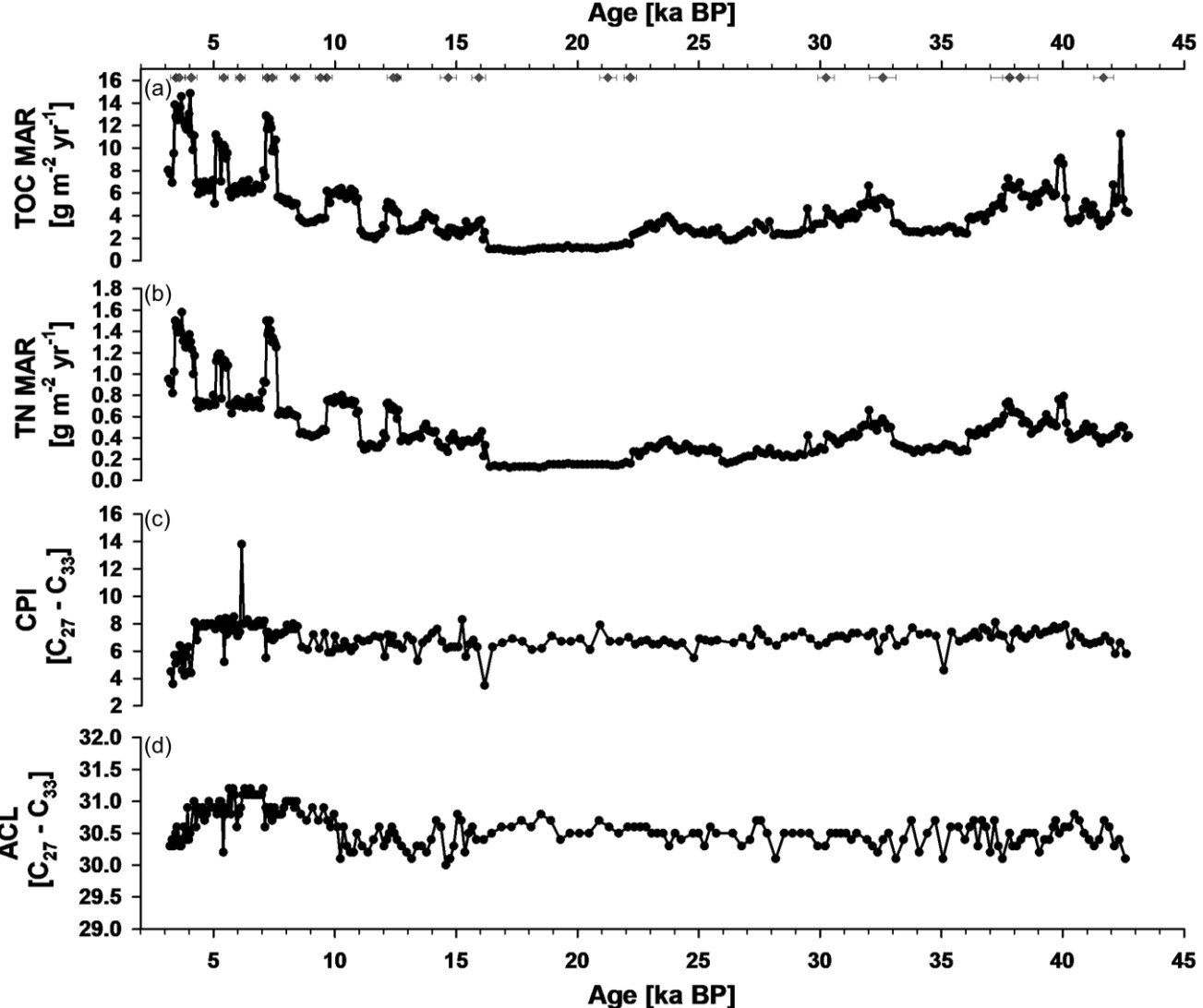

**Fig. A1:** Bulk and plant-wax derived *n*-alkane data of SL167. (a) Total mass accumulation rates of total organic carbon (TOC MAR), (b) total mass accumulation rates of total nitrogen (TN MAR), (c) carbon preference index (CPI) of the *n*-alkanes $C_{27}$ to $C_{33}$ and (d) average chain length (ACL) of the *n*-alkanes $C_{27}$ to $C_{33}$. Diamonds showing dated ages of SL167.

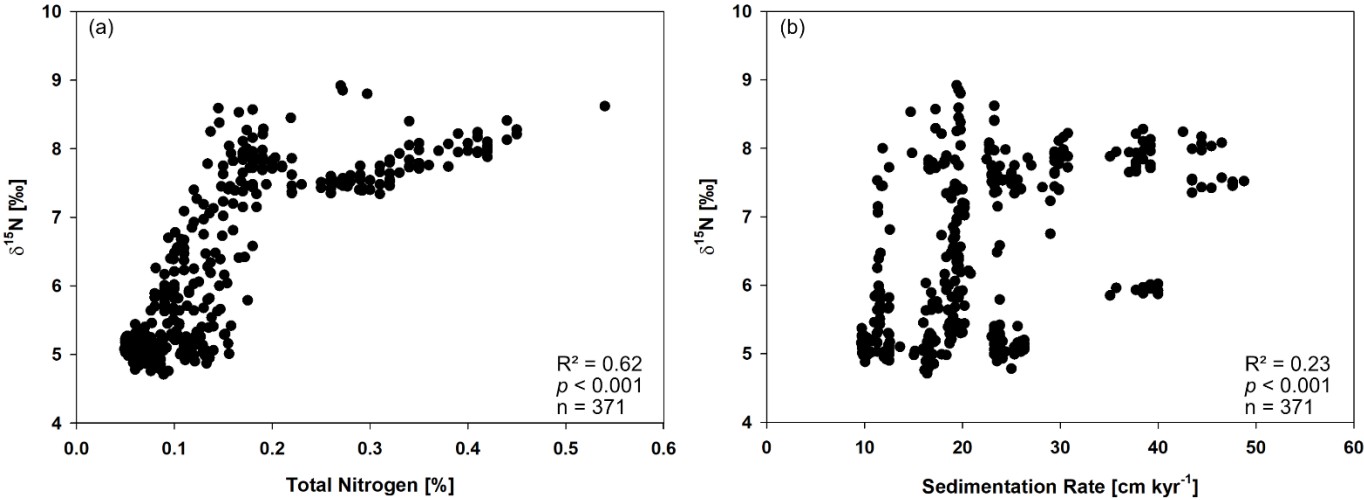

**Fig. A2:** Crossplots of (a) total nitrogen (TN) and δ¹⁵N as well as (b) the sedimentation rate and δ¹⁵N of sediment core SL167.

## Data availability

The data sets are stored and available at PANGAEA under https://doi.pangaea.de/10.1594/PANGAEA.964226 and https://doi.org/10.1594/PANGAEA.960060 (census date of benthic foraminifera).

## Author contributions

NB: conceptualization, formal analysis, investigation, methodology, visualization, writing – original draft preparation. GS: conceptualization, faunal analysis, investigation, methodology, visualization, resources, supervision, writing – original draft preparation. BG: conceptualization, writing – original draft preparation. HS: faunal analysis, writing – original draft preparation. PM: faunal analysis, writing – original draft preparation.

## Competing interests

The contact author has declared that none of the authors has any competing interests.

**Acknowledgements**

This work is funded by the Deutsche Forschungsgemeinschaft (DFG, German Research Foundation) under Germany's Excellence Strategy – EXC 2037 'CLICCS - Climate, Climatic Change, and Society' – Project Number: 390683824, contribution to the Center for Earth System Research and Sustainability (CEN) of Universität Hamburg. Chlorophyll a analyses and visualizations used in this paper were produced with the Giovanni online data system, developed and maintained by the NASA GES DISC. We also acknowledge the MODIS mission scientists and associated NASA personnel for the production

of the data used in this research effort. We thank Frauke Langenberg, Marc Metzke, Miriam Warning, Jan Maier, Dorothea Bunzel, Tobias Winkler and Sabine Beckmann for technical and analytical support. We also thank the two anonymous reviewer for their constructive comments and improving the manuscript.

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

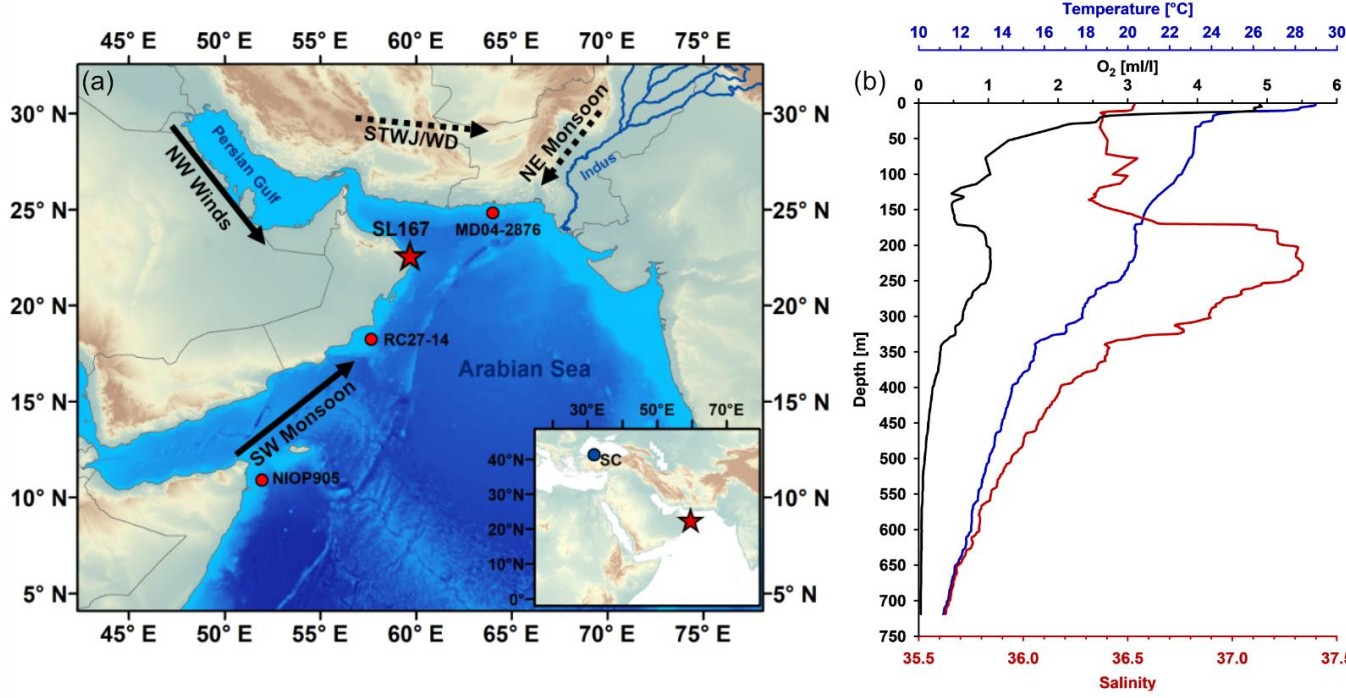

**Figure 1: (a) Map of the study site SL167 (red star), nearby marine records (red circles) MD04-2876 (Pichevin et al., 2007), RC27-14 (Altabet et al., 2002), NIOP905 (Ivanochko et al., 2005) and stalagmite record (blue circle in inset map) Sofular Cave (SC)(Fleitmann et al., 2009). The black arrows (SW monsoon & NW winds) show the prevailing wind pattern during the summer and the dashed black arrows (NE monsoon & subtropical westerly jet/western disturbances (STWJ/WD)) indicate the prevailing wind pattern during the winter (after Clift and Plumb, 2008; Hunt et al., 2018). Map was created using ArcGIS v.10.8 (ESRI, 2019). The bathymetric data are from the General Bathymetric Chart of the Oceans (GEBCO, 2014; http://www.gebco.net, last access: 4 January 2017). (b) Temperature (blue), oxygen (O₂, black) and salinity (red) depth profile obtained at the end of the SW-monsoon season in September 2007 during M74/1b cruise at the location of SL167.**



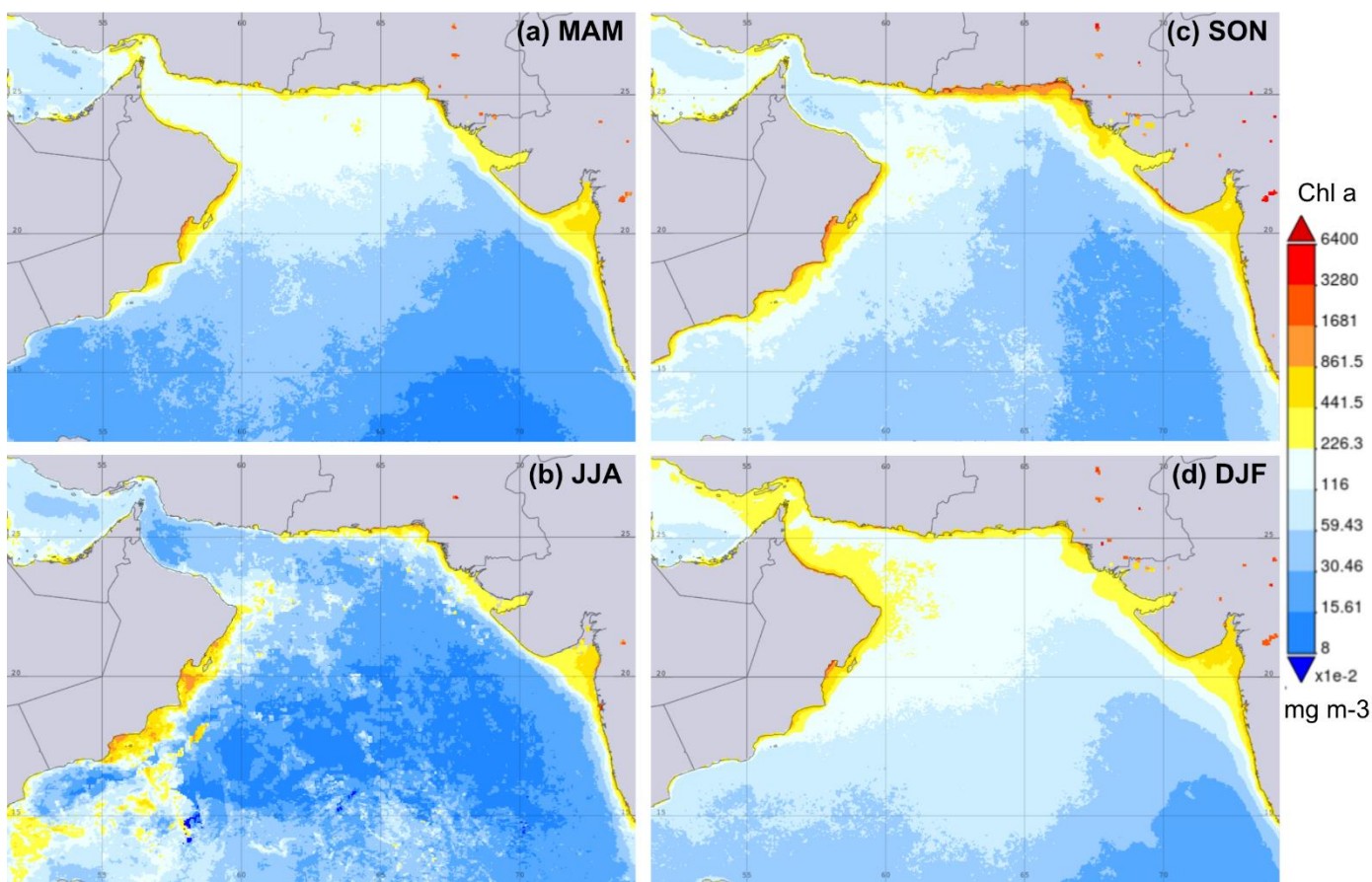

**Figure 2: Average chlorophyll a concentration (MODIS Aqua, L3m, 4 km spatial resolution, monthly) for the different seasons (a) March, April, May (MAM), (b) June, July, August (JJA), (c) September, October, November (SON) and (d) December, January, February (DFJ) over the period from 2002 to 2022 (NASA, 2022). Data were visualized using the Giovanni online data system, developed and maintained by the NASA GES DISC (date of access: 01/26/2023).**


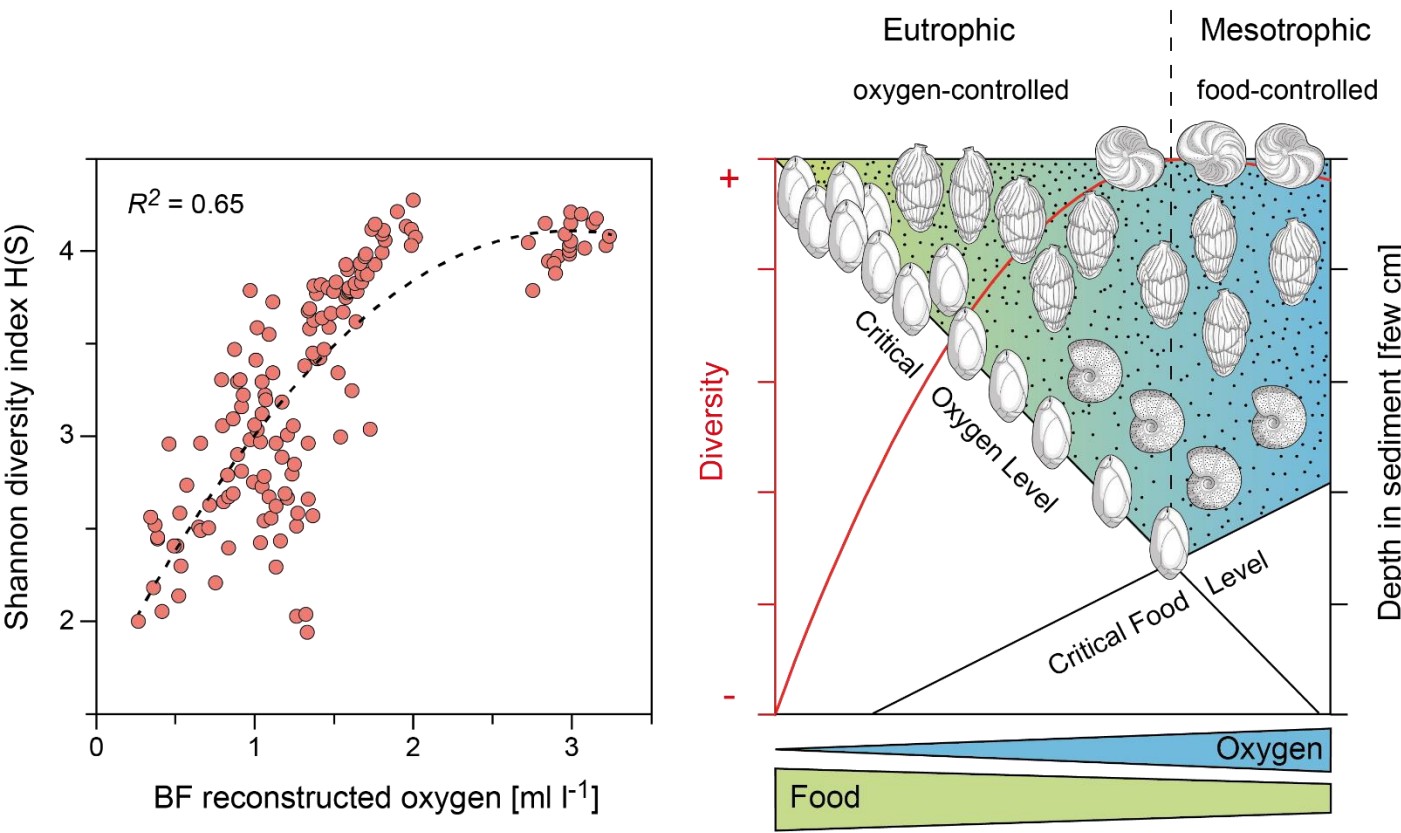

**Figure 3: Right:** Eutrophic to mesotrophic section of the Trophic-Oxygen (TROX) model of Jorissen et al. (1995) with addition of Gooday (2003) describing the general dependence of benthic foraminiferal microhabitat structure and diversity from oxygen concentration and food supply. Figure modified from Schmiedl et al. (2023). **Left:** Reconstructed dissolved oxygen concentration of bottom and pore waters versus benthic foraminiferal diversity H(S) for core SL167. The stippled line represents a polynomial of 2nd degree model with p < 0.001. The observed significant correlation suggests a dominant oxygen control of benthic foraminiferal diversity at site SL167.

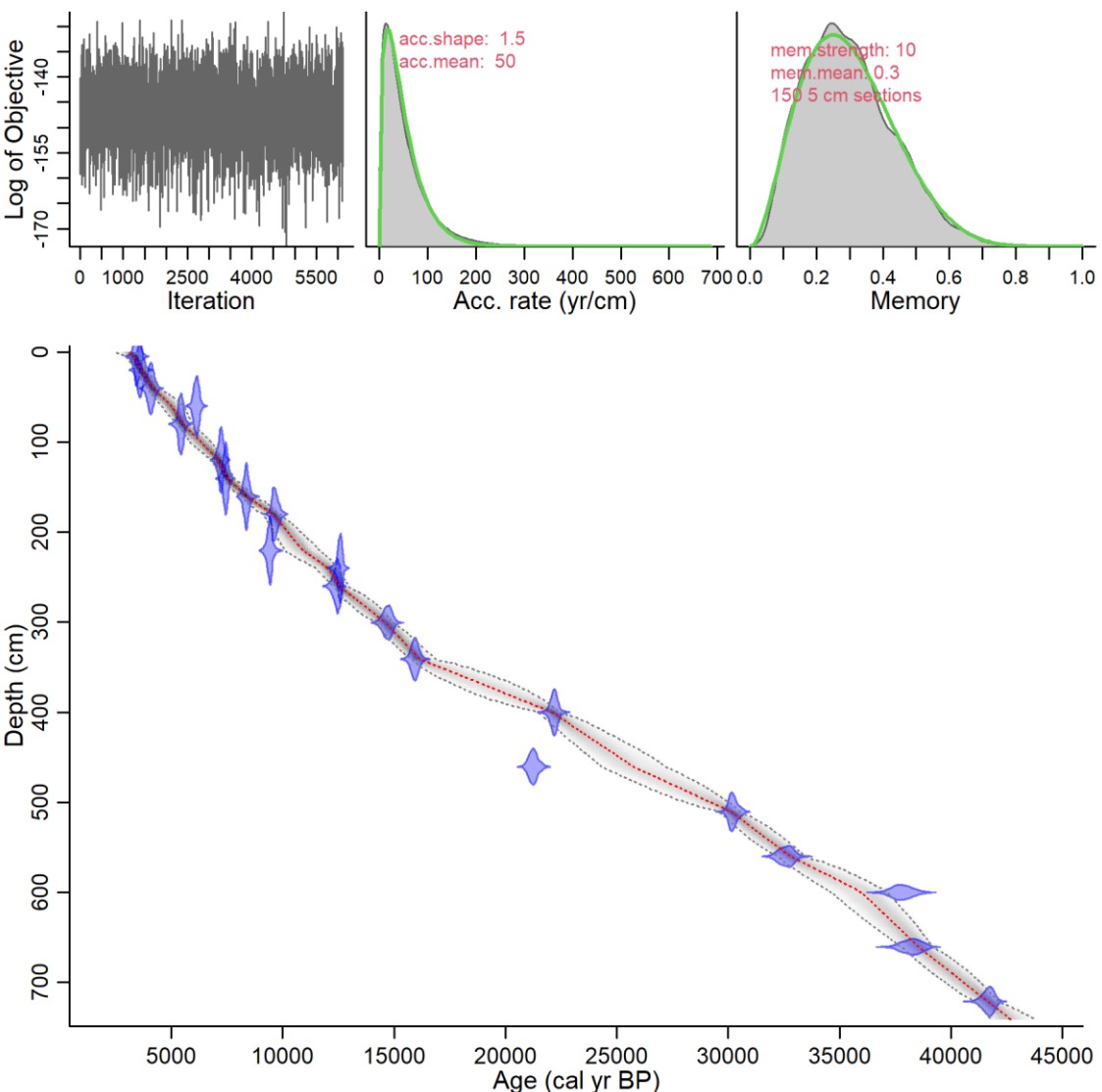

**Figure 4: Age-depth model of SL167 using the R package BACON v. 2.5.6 (Blaauw and Christen, 2011). The upper panels show the Markov Chain Monte Carlo iterations (left), the distributions of the prior (green curve) and posterior (grey area) accumulation rates (middle) and memory (right). The lower panel shows the age-depth model of SL167. The calibrated $^{14}$C dates are shown in blue. The red line shows the modelled mean age of SL167 with the 95 % confidence interval (black dotted lines). A ΔR of 93 ± 61 years was applied.**

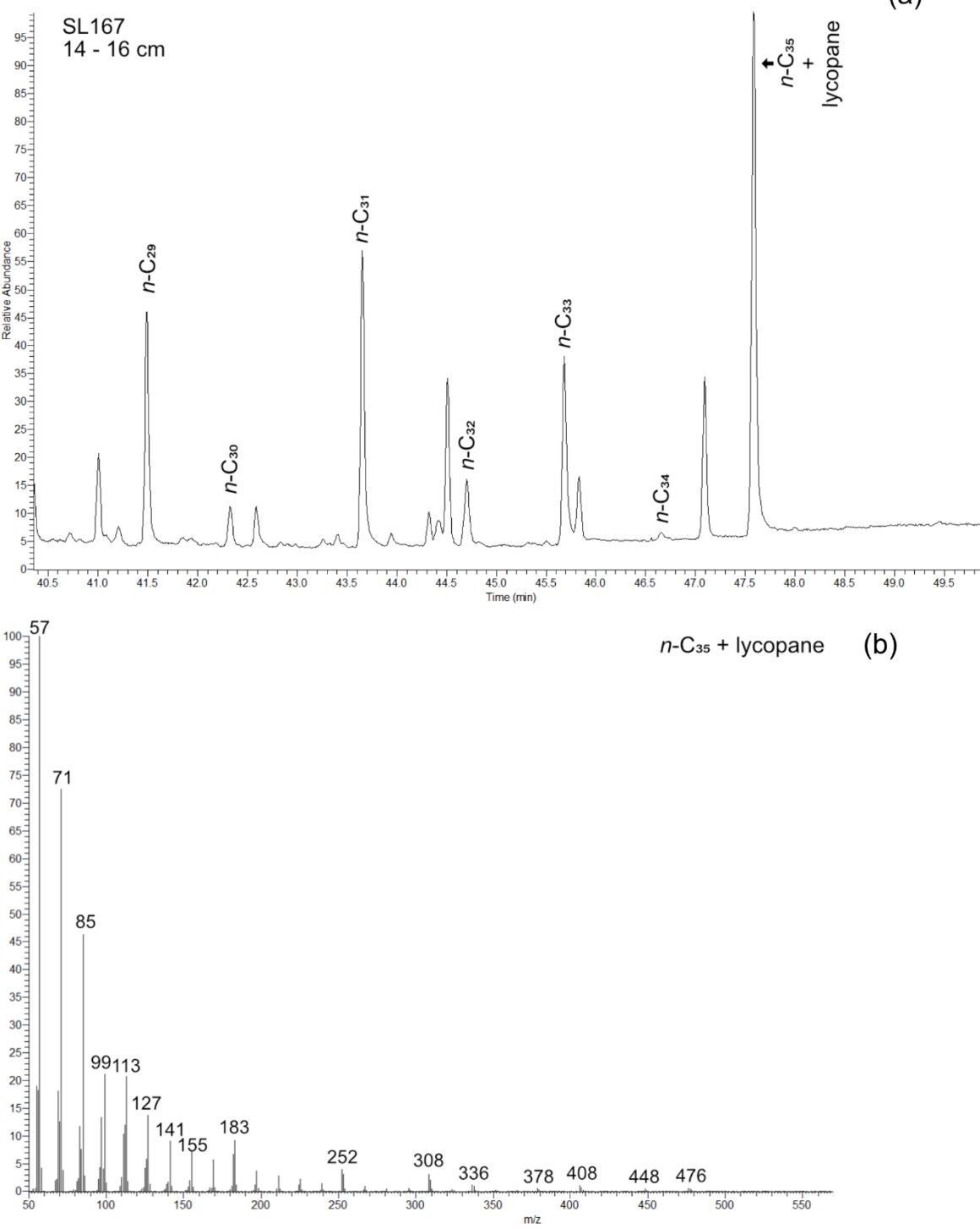

**Figure 5: GC-MS measurement of the apolar fraction of sample 14-16 cm with retention time (a) and mass spectra of the *n*-alkane C$_{35}$ + lycopane (b).**

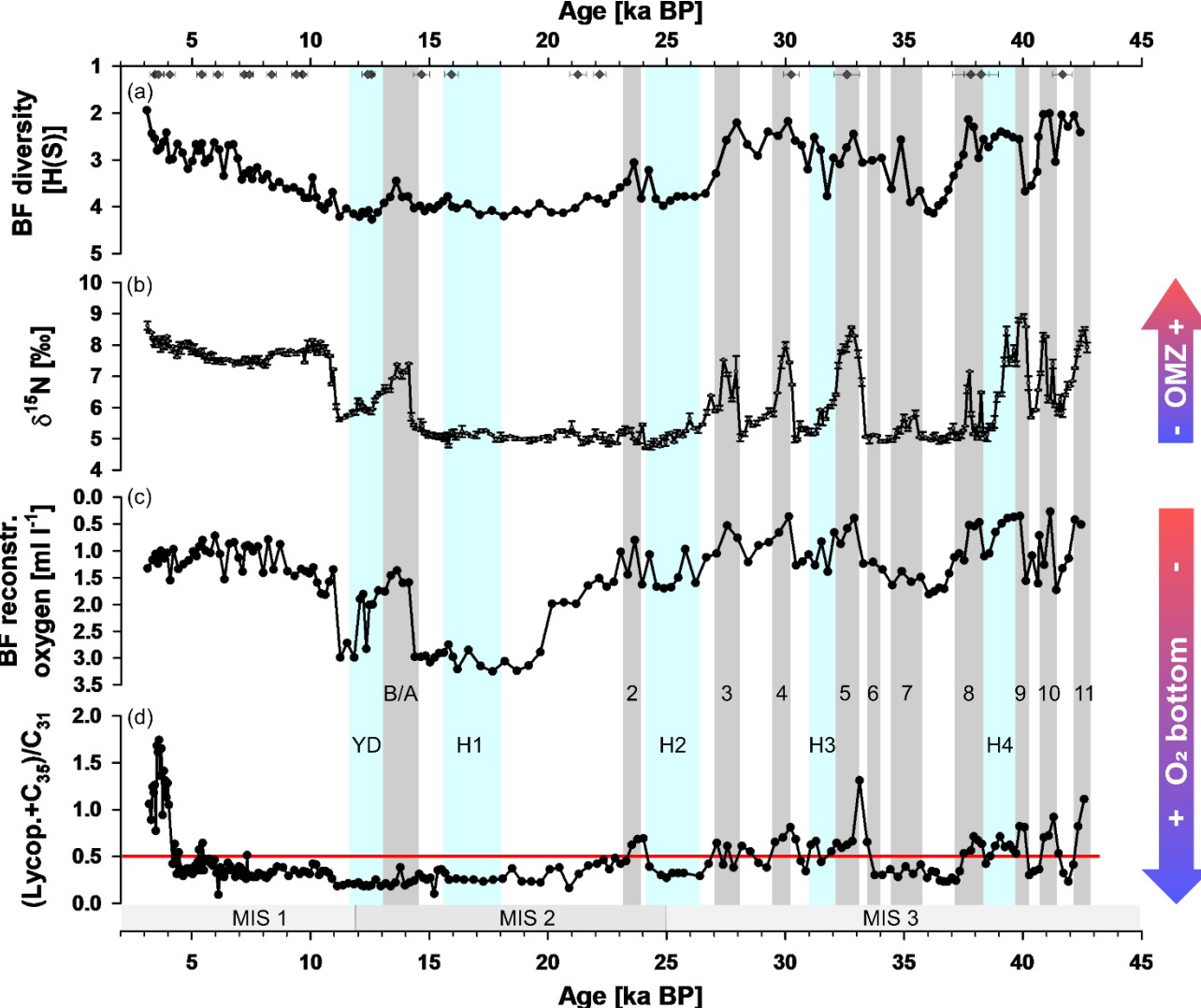


**Figure 6: Oxygen proxies of SL167.** Shannon Wiener (H(S)) diversity index for benthic foraminifera (BF, a), nitrogen isotope values (δ$^{15}$N) as denitrification and oxygen minimum zone (OMZ) strength indicator for the water column (b), reconstructed bottom water oxygen content by benthic foraminifera (c) and ratio of (lycopane + $n$-C$_{35}$)/$n$-C$_{31}$ as indicator for bottom water oxygen (d). The red line in (d) indicates the threshold of oxic to sub-/anoxic conditions (Sinninghe Damsté et al., 2003). Grey bars indicate Bølling/Allerød (B/A) and Dansgaard-Oeschger events (D/O, numbers) mostly based on Fleitmann et al. (2009). Blue bars indicate Younger Dryas based on Fleitmann et al. (2009) and Heinrich (H) events based on a compilation by Allard et al. (2021). Diamonds showing dated ages of SL167. Note the reversed axes for H(S) and BF reconstructed oxygen. The grey bars at the bottom represents the different Marine Isotope Stages (MIS).

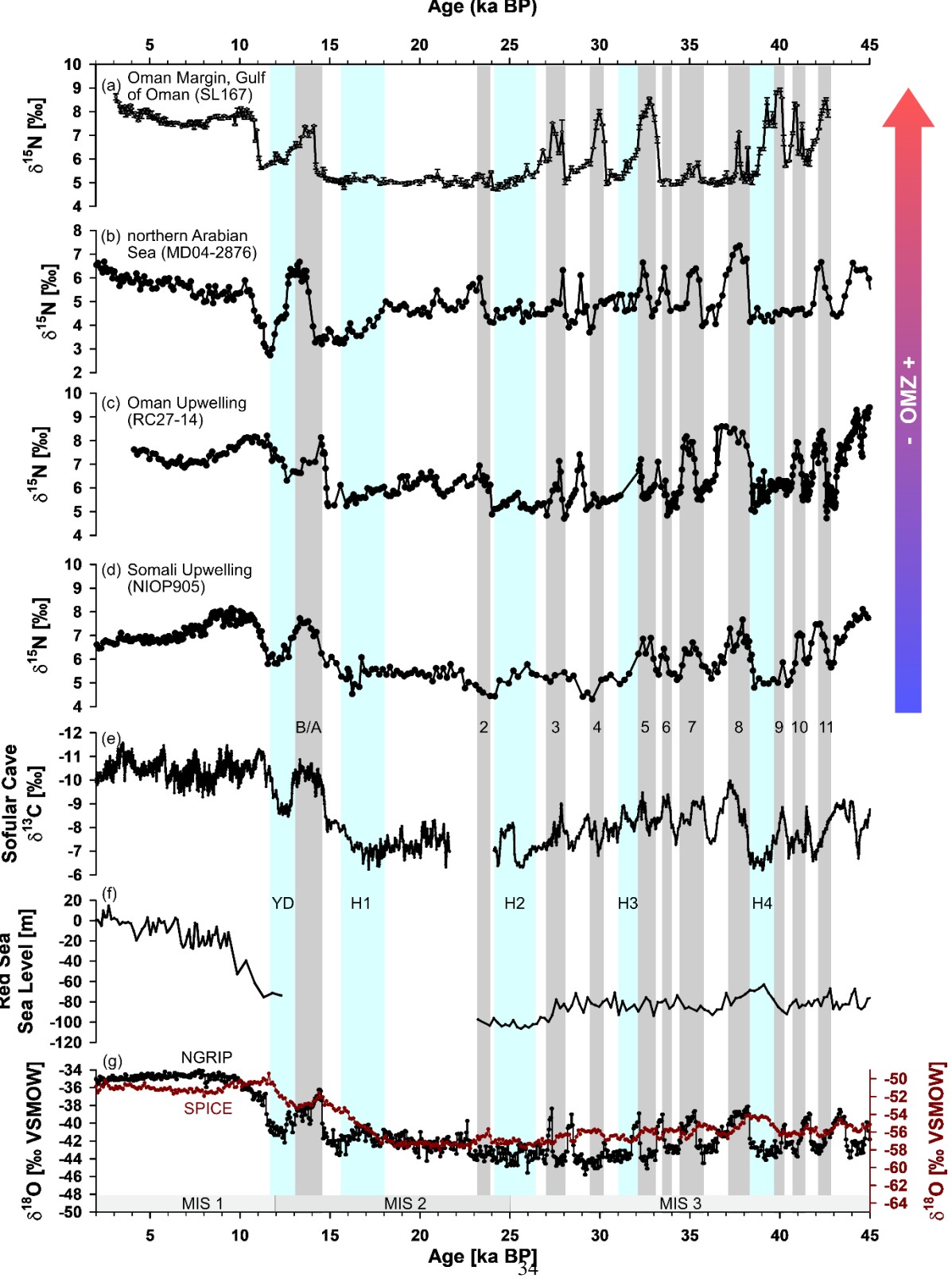

**Figure 7: Comparison of high-resolution nitrogen isotope ($\delta^{15}$N) records in the Arabian Sea. SL167 (this study) from the Gulf of Oman (a), MD04-2876 (Pichevin et al., 2007) from the Pakistan margin in the northern Arabian Sea (b), RC27-14 (Altabet et al., 2002) from the Oman Upwelling region in the western Arabian Sea (c) and NIOP905 (Ivanochko et al., 2005) from the Somali Upwelling in the southwestern Arabian Sea (d). Grey bars indicate Bølling/Allerød (B/A) and Dansgaard-Oeschger events (D/O, numbers) mostly based on the $\delta^{13}$C Sofular cave record (e) by Fleitmann et al. (2009). Combined Red Sea Sea Level reconstruction**

**of sediment core KL11 (f) by Siddall et al. (2003) and Rohling et al. (2008) for the Holocene and Pleistocene parts, respectively. The $\delta^{18}$O records of NGRIP (North Greenland Ice Core Project members, 2004) and SPICE (Steig et al., 2021) are shown in (g). Blue bars indicate Younger Dryas based on Fleitmann et al. (2009) and Heinrich (H) events based on a compilation by Allard et al. (2021). The grey bars at the bottom represents the different Marine Isotope Stages (MIS).**

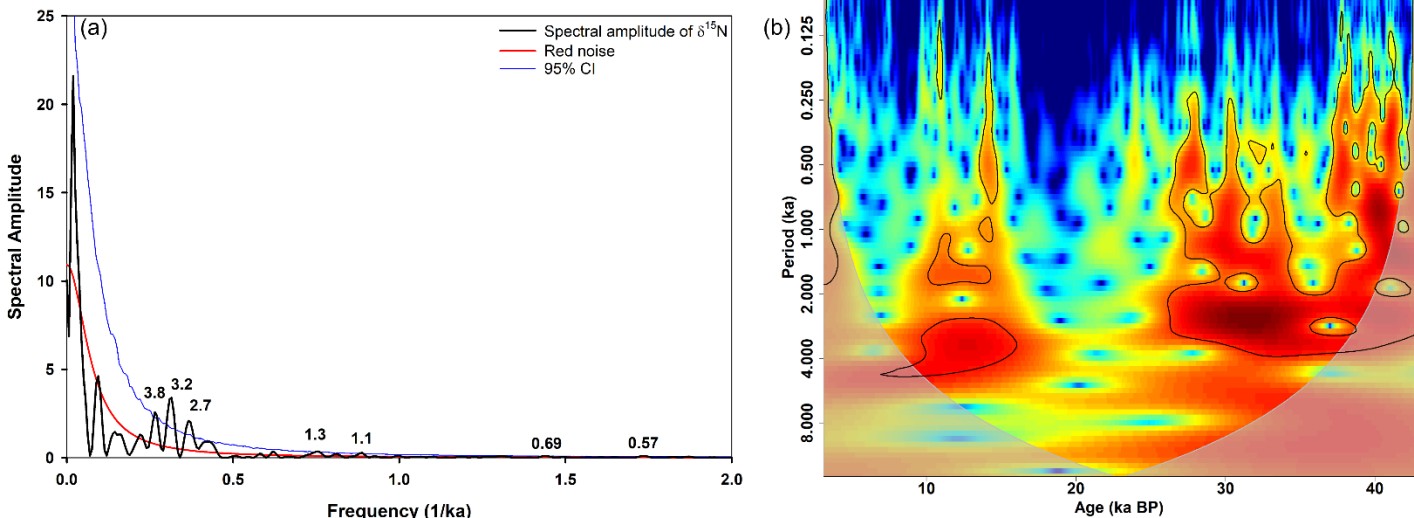

**Figure 8: Spectral analysis (a)) and wavelet power spectrum (b)) of δ¹⁵N of SL167. Red and blue lines in (a) indicate the red noise level and 95 % confidence interval (CI), respectively. Black numbers show significant periodicities on the 95 % CI. The black lines in (b) indicates the 95 % significance level. The colours in b) represent the power of the wavelet power spectrum, with red (blue) showing high (low) power. The grey shaded area represents the cone of influence.**

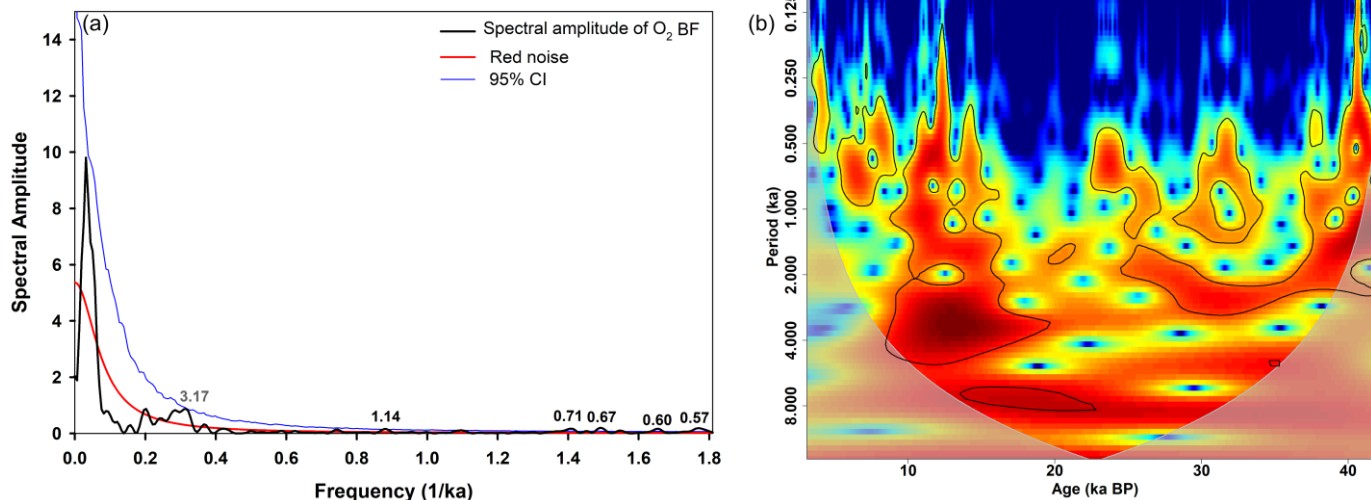

**Figure 9: Spectral analysis (a)) and wavelet power spectrum (b)) of the reconstructed oxygen content of SL167 by using benthic foraminifera assemblage (O₂ BF). Red and blue lines in (a) indicate the red noise level and 95 % confidence interval (CI), respectively. Black numbers show significant periodicities on the 95 % CI, the grey number (3.17 ka) the significant periodicity on the 90 % CI. The black lines in (b) indicates the 95 % significance level. The colours in b) represent the power of the wavelet power spectrum, with red (blue) showing high (low) power. The grey shaded area represents the cone of influence.**

**Tables:**


**Table 1: AMS $^{14}$C dates of planktic foraminifera of sediment core SL167 and calibrated ages using Calib 8.2 with Marine20 calibration curve and ΔR of 93 ± 61 years.**

| Depth interval (cm) | Mean depth (cm) | Material | $^{14}$C age (yr BP) | Calendar age 95,4 % prob. (yr BP) | Calendar age max (yr BP) | Calendar age min (yr BP) |
|---|---|---|---|---|---|---|
| 4.5–5.5 | 5 | Planktic foraminifera | 3790 ± 30 | 3457 ± 216 | 3673 | 3241 |
| 19.5–20.5 | 20 | Planktic foraminifera | 3910 ± 30 | 3604 ± 217 | 3821 | 3387 |
| 40–41 | 40.5 | Planktic foraminifera | 4280 ± 30 | 4083 ± 241 | 4323 | 3842 |
| 59.5–60.5 | 60 | Planktic foraminifera | 6000 ± 30 | 6112 ± 194 | 6305 | 5918 |
| 79.5–80.5 | 80 | Planktic foraminifera | 5350 ± 30 | 5425 ± 193 | 5618 | 5232 |
| 119.5–120.5 | 120 | Planktic foraminifera | 7000 ± 30 | 7216 ± 196 | 7412 | 7020 |
| 140–141 | 140.5 | Planktic foraminifera | 7230 ± 30 | 7432 ± 171 | 7602 | 7261 |
| 160–161 | 160.5 | Planktic foraminifera | 8160 ± 30 | 8364 ± 192 | 8556 | 8172 |
| 179–181 | 180 | Planktic foraminifera | 9200 ± 30 | 9673 ± 225 | 9897 | 9448 |
| 220–221 | 220.5 | Planktic foraminifera | 9010 ± 30 | 9410 ± 202 | 9612 | 9208 |
| 239.5–240.5 | 240 | Planktic foraminifera | 11300 ± 30 | 12576 ± 176 | 12737 | 12386 |
| 259–261 | 260 | Planktic foraminifera | 11170 ± 30 | 12413 ± 241 | 12654 | 12172 |
| 299–302 | 300.5 | Planktic foraminifera | 13080 ± 40 | 14679 ± 345 | 15024 | 14334 |
| 340.5–341.5 | 341 | Planktic foraminifera | 13980 ± 40 | 15936 ± 284 | 16219 | 15652 |
| 399–401 | 400 | Planktic foraminifera | 19200 ± 70 | 22178 ± 263 | 22441 | 21915 |
| 460–461 | 460.5 | Planktic foraminifera | 18430 ± 50 | 21260 ± 350 | 21609 | 20910 |
| 510–511 | 510.5 | Planktic foraminifera | 26950 ± 120 | 30252 ± 338 | 30590 | 29914 |
| 559.5–560.5 | 560 | Planktic foraminifera | 29260 ± 140 | 32593 ± 539 | 33132 | 32054 |
| 599–601 | 600 | Planktic foraminifera | 33930 ± 230 | 37818 ± 777 | 38595 | 37041 |
| 660–661 | 660.5 | Planktic foraminifera | 34260 ± 220 | 38255 ± 720 | 38974 | 37535 |
| 720–722 | 721 | Planktic foraminifera | 38290 ± 320 | 41684 ± 415 | 42098 | 41269 |