# Peer review of "Distinct oxygenation modes of the Gulf of Oman during the past 43,000 years – a multi-proxy approach"

_EGUsphere, 2023_

## Author Comment (AC1)

**Comments Anonymous Referee #1**

Burdanowitz and co-authors present a study exploring the OMZ dynamics in the Gulf of Oman across the past 43,000 years. They employ a multi-proxy approach, using bulk sedimentary nitrogen isotopes, n-alkane ratios, and benthic foraminiferal analysis to reconstruct Oxygen Minimum Zones and bottom water oxygen conditions during different climatic periods. From this, the authors gain insights into varied oxygenation modes of the Gulf of Oman, delineating stable periods characterised by either pronounced OMZ or well-oxygenated conditions and unstable phases exhibiting oscillating oxygenation.

The manuscript is well written. The data is well documented and presented. The references are up-to-date.

While the paper provides valuable insights into the oxygenation dynamics of the Gulf of Oman (and Arabian Sea), there are some potential limitations and caveats to consider:

The main points of the study come from the d15N records, thus a more in-depth discussion of the robustness of the proxy would be good. For example the authors could show total organic nitrogen in a supplementary plot as a time series, but also as a cross plot with d15N to make sure none of the signal is driven by preferential degradation/ secondary overprints in the water column.

How do the authors know that d15N was a local signal and was not transported from further south/ further north? How do the core sites nitrate d15N look like nowadays? Maybe check Martin and Casciotti, 2017. Is the offset between the sites also observed/maintained in the past? Could a changing offset mean less/more water mass transport/mixing of d15N signal or is it all attributed to local changes in water column denitrification (in connection with O2)?

Response: First, we thank the referee for her/his comments and suggestions, which helps us to improve the manuscript.

To address the robustness and limitations of $\delta^{15}N$ as proxy we will add a paragraph to the introduction as follows: "Denitrification occurs in oxygen-depleted oceanic zones, serving as oxygen source for heterotrophic bacteria reducing nitrate via nitrite to $N_2$ (Devol, 2008; Rixen et al., 2020 and references therein). For these heterotrophic bacteria it is energetically more efficient to use the lighter $^{14}N$ from the nitrate source (Devol, 2008). Because of this isotopic fractionation, the remaining nitrate, which is used by phytoplankton, becomes more enriched in $^{15}N$ (Montoya, 2008 and references therein). The global average $\delta^{15}N$ of deep water nitrate is about 4.8 ± 0.2 ‰ (Sigman et al., 2000) but denitrification processes lead to more enriched $\delta^{15}N$ in nitrate with values above 18 ‰ in the Arabian Sea OMZ (Rixen et al., 2014). During the nitrate consumption, phytoplankton also uses the lighter $^{14}N$ and have an isotope discrimination factor of about 5 ‰ and incorporate the denitrification signal into the biomass (Montoya, 2008). Sinking of this biomass to the ocean floor transports the denitrification signal to the sediments (e.g. Altabet et al., 1995; Gaye-Haake et al., 2005). Therefore, $\delta^{15}N$ values can be used as a proxy for the OMZ strength in the water column (e.g. Altabet et al., 1995; Reichart et al., 1997). However, diagenetic processes can alter sedimentary δ15N values especially under low sedimentation rates (Gaye-Haake et al., 2005; Jung et al., 1997; Junium et al., 2015; Möbius et al., 2011; Tesdal et al., 2013). Nevertheless, it was found that in the Arabian Sea, especially under high sedimentation rates, sedimentary $\delta^{15}N$ values are a reliable indicator for past denitrification processes (Möbius et al., 2011)."

We did not measure the total organic nitrogen but total nitrogen and will add the mass accumulation rates (MAR) of total nitrogen, with similar behavior to TOC MAR, to Fig. A1 in the appendix. We also will add cross plots of TN- $\delta^{15}N$ and sedimentation rates – $\delta^{15}N$, showing in general more enriched

$\delta^{15}N$ with higher TN content ($R^2 = 0.62$, $p < 0.001$, $n = 371$), but no trend/weak correlation of sedimentation rates – $\delta^{15}N$ ($R^2 = 0.23$, $p < 0.001$, $n = 371$). Even the p-value argue for a significant correlation, $R^2$ is low and the number of samples is high, leading to this p-value. The sedimentation rates in core SL167 range between 9.7 and 49 cm/kyr and are high enough for an unaltered signal of $\delta^{15}N$ (see Jung et al., 1997 and Möbius et al.,2011). The correlation between TN and $\delta^{15}N$ may be to a higher productivity and/or better preservation of organic matter under sub-/anoxic conditions. We will discuss this in the discussion part.

There are no data available for modern nitrate $\delta^{15}N$ at the core site. Gaye et al., 2013 found subsurface nitrate $\delta^{15}N$ values of about 7 to 14 ‰ to the south in the Oman Upwelling area. The data set by Martin & Casciotti (2017) covers only the central and eastern Arabian Sea. However, surface sediments from across the Arabian Sea show bulk $\delta^{15}N$ values between 6 and 12 ‰ with most enriched $\delta^{15}N$ in the central part and also higher values (8 – 10 ‰) along the Oman margin (Gaye et al. 2018) matching the late Holocene $\delta^{15}N$ values of SL167. It is also possible that the $\delta^{15}N$ can be affected by changes in the water mass transport/mixing or lateral transport, e.g. from Oman Upwelling area. We will add this to the discussion.

**Minor comments:**

Text:

Line 11-12: This sentence doesn't make sense. Do you mean: The understanding of the dynamics of the OMZ, its marine environmental is of importance due to its climate feedbacks.

Response: We will rewrite the sentence to "It is important to understand the dynamics of the OMZ and related marine environmental conditions because of its important climate feedbacks".

Line 14: "for the first time": oxygenation has been reconstructed in previous studies, what part of this is novel?

Response: The novelty of this study is, that we apply the multi-proxy approach of the oxygen status of the water column ($\delta^{15}N$) and bottom water ((lycopane + n-$C_{35}$)/n-$C_{31}$ & benthic foraminiferal faunal analysis) for the first time in the Gulf of Oman, lying in the "shadow" of the Oman Upwelling region. We will add further explanation to its novelty: "This multi-proxy approach is done for the first time at the northeast Oman margin in the Gulf of Oman.".

Line 15: "bulk sedimentary nitrogen isotopes"

Response: We will add "bulk sedimentary".

Line 23: Mention the proposed mechanism for the unstable period of oscillating oxygenation

Response: We will add the proposed mechanism to abstract as follows: "The unstable period may be triggered by an interstadial AMOC mode, which is required to initiate D/O events"

Line 29: N20 -> N2O

Response: We will change the typo.

Line 34: how does it affect the diversity? making it more/less diverse?

Response: In general, low oxygen conditions results in lower diversity since heterotrophic organisms rely on oxygen (e.g. Helly & Levin, 2004). However, compared to other non-OMZ regions, the diversity is still comparatively high due to their high variety of infaunal benthic foraminifera, which are well adapted to low oxygen conditions (Schmiedl et al., 2023). We will add following to the manuscript: "Although the diversity of benthic foraminifera is generally reduced under low oxygen conditions, the OMZ faunas of the Arabian Sea seem to be more resilient to low oxygen conditions when compared to other non-OMZ regions. This higher resilience can be attributed to a dominance of nitrate respiring infaunal benthic foraminifera in OMZ regions (Schmiedl et al., 2023)."

Line 41: what does a "dry" Persian Gulf mean —> exposed due to lower sea level?

Response: With the term "dry" Persian Gulf, we mean the drying and exposure of the basin due to lower sea level. Further, due to the lower sea level, the connection (Strait of Hormuz) is closed and no ventilation via the PGW can occur. We will explain this part in more detail and change it to: "The authors also concluded that a strengthening of the OMZ in the Gulf of Oman can be also induced by lower sea level. In this scenario, the Persian Gulf has fallen dry and, therefore, the ventilation via the PGW would stop completely (Lachkar et al., 2019).".

Line 57: Add a sentence about why the d15N is elevated with denitrification. Use a more denitrification/d15N specific reference, e.g. Brandes et al., 1998, Cline and Kaplan, 1975 or review by Capone et al., 2008

Response: We will rewrite the whole paragraph adding further information about the usage and limitation of $\delta^{15}N$ as proxy as written in the comment above regarding the main issues.

Line 74: strictly speaking the location is at the rim of the Gulf of Oman, why not calling it Oman margin? Can you show somewhere that hydrographically the Gulf of Oman (rim) is distinct from the Oman margin.

Response: The core is located at the continental slope west of Ra's al Hadd, with the latter separating the Gulf of Oman from the Arabian Sea (UNEP, 1999) to the south. To the east, the Gulf of Oman is separated from the Arabian Sea by the Murray Ridge (e.g. Shimmield et al., 1990, Szuman et al., 2006, Uchupi et al., 2002). It is true, that the core location belongs to the Oman margin (e.g. Szuman et al., 2006, Uchupi et al., 2002), like the region of the Oman Upwelling area in the Arabian Sea. However, we want to highlight that the core site is, in term of its oceanographic/geographical setting, located in the Gulf of Oman and in the "shadow region" of the Oman Upwelling area offshore eastern Oman. Nevertheless, we are aware about the concerns of the referee and agree to specify the location of the core. To make it clear we will add "northeast Oman margin continental slope within the Gulf of Oman". We will also add this to the abstract and the material and methods section.

Line 86: Based on their depths, wouldn't the RSW also count as intermediate waters?

Response: Yes, the Red Sea Water is also an intermediate water mass. We will add this information to this section.

Lines 91-99: when talking about productivity and seasons add the panels in refs for Figure 2. e.g. "In contrast, highest productivity in the northern Arabian Sea and Gulf of Oman is observed during the winter season (Figure 2d).

Response: We agree that it is useful to add the panels of Figure 2 for the mentioned summer (Figure 2b) and winter season (Figure 2d), respectively. We will change this.

Line 110: which planktonic foraminifera?

Response: We used the surface-dwelling planktic foraminifera *Globigerinoides spp.*, *Globigerinella spp.* and *Orbulina universa* for the radiocarbon dating. We will add this information.

Line 145: add references for ACL and CPI.

Response: We thank the referee pointing out this issue. Hence, the ACL and CPI are only side aspects of this manuscript, and we realized that there is not enough information about these proxies and references are missing. We will add further basic information and references in the method section as follows: "The average chain length (ACL) of plant-wax derived $n$-alkanes are commonly used to identify plant functional types and environmental conditions (e.g. Collister et al., 1994; Cooper et al., 2015; Eglinton and Eglinton, 2008; Rommerskirchen et al., 2006). For instance, plants located in arid environments tend to have longer ACL than plants in humid environments (e.g. Carr et al., 2014; Rommerskirchen et al., 2006; Vogts et al., 2009). But the ACL also differs with the plant functional type, for instance $C_4$ grasses have in general higher ACL than woody gymnosperms (e.g. Bush and McInerney, 2013; Carr et al., 2014; Cooper et al., 2015). However, the validity of the ACL is limited and, if possible, should be combined with compound-specific isotope measurements of the $n$-alkanes (e.g. Eglinton and Eglinton, 2008; Rommerskirchen et al., 2006; Vogts et al., 2009)." and "The carbon preference index (CPI), an indicator for the odd-over-even predominance, is usable to distinguish between terrestrial plant and petroleum sources (e.g. Bray and Evans, 1961; Cranwell, 1978, 1981; Pancost and Boot, 2004). Terrestrial plant derived $n$-alkanes and recent sediments with unaltered organic material have an odd-over-even predominance with a CPI higher than 3 (e.g. Bray and Evans, 1961). In contrast, petroleum sources have higher abundance of even $n$-alkanes with CPIs < 1 and are an indication of higher degradation (e.g. Bray and Evans, 1961)."

Line 155: add how oxygen concentration would affect diversity index.

Response: We will add further information to this section as follows: "The H(S) considers the number of species and their relative proportion in the sample. The H(S) value is at a maximum, when all species have equal proportions, while species with low abundances contribute little to it. In eutrophic to mesotrophic ecosystems, the diversity and microhabitat structure is oxygen-controlled, as predicted by the TROX (Trophic-Oxygen) model (Gooday, 2003; Jorissen et al., 1995) (Figure 3).". In addition, we will add a figure (will be Figure 3) showing the relationship of the diversity index, oxygen and food availability.

Line 175: The authors initially discuss the lipid biomarker results, thus I would suggest to flip the order in the methods section (first describe lipid biomarkers and then d15N).

Response: We agree with the referee. We will change the order of the lipid biomarker and $\delta^{15}N$ section in the methods to be consistent.

Line 185: also add a results section about the long-term trend of lycopane and refer to Fig. 5d

Response: We agree and will add a section about the long-term trend of lycopane as follows: "The (lycopane + $n$-C$_{35}$)/$n$-C$_{31}$ varies between 0.1 and 1.7 throughout the core (Figure 5d). Highest ratios occur during the late Holocene (3 – 4 ka BP) and warmer Pleistocene periods reconstructed by the NGRIP ice core $\delta^{18}$O (North Greenland Ice Core Project members, 2004) and Sofular cave $\delta^{13}$C (Fleitmann et al., 2009) records, respectively." Note that the number of the figures will change due to the insert of a new figure.

Line 187: did you also quantify TON to test the robustness of d15N?

Response: We measured and quantified the total nitrogen content of the sediments but not total organic nitrogen prior to $\delta^{15}$N measurements. We will add the total nitrogen mass accumulation rates to Fig. A1 and cross plots of TN- $\delta^{15}$N and sedimentation rates – $\delta^{15}$N in the appendix. As written above (reply to main point), there is a possibility of a diagenetic overprint of the $\delta^{15}$N signal. However, the sedimentation rates are high at the core site (9.7 to 49 cm/kyr) and from other studies (e.g. Möbius et al., 2011 and Jung et al., 1997) we know, that with this high sedimentation rates it is unlikely that $\delta^{15}$N is altered by diagenesis. We will add this to the discussion.

Line 201+: This section would benefit from a few subtitles to structure the discussion a bit better and to help the reader to follow the argumentation.

Response: We agree with the referee and we will restructure this section. We will split this part into two main sections. The first one for the time period Pleistocene until Younger Dryas, with own paragraphs for the LGM and B/A and the other main section for the Holocene.

Line 205: D/O 6 and 7 might not be observed in bottom water reconstruction due to lower resolution?

Response: It is true that the resolution is a bit lower during the Pleistocene than the Holocene part. However, for DO 6 (DO 7) we have 2 (4) data points for the ratio of (lycopane + C$_{35}$)/C$_{31}$ and 1 (4) for the benthic foraminifera data set, respectively. This is a similar resolution like for DO 2 – 5 and DO 9 – 11. Further, the $\delta^{15}$N (proxy for the water column oxygenation) also shows a weaker OMZ during that time. Both proxies for bottom water oxygenation are independent from each other and together with $\delta^{15}$N we believe that this is a reliable signal during DO 6 and 7.

Line 205: "most intense" in what direction? Maybe choose another word for this?

Response: We will change this to "Highest reconstructed bottom water oxygen content"

Line 206: superscript -1

Response: We will change this.

Line 206: highlight Younger Dryas with a bar in Fig. 5.

Response: We will add a light blue bar to highlight the Younger Dryas in this figure (now 6) and also the next one (now 7).

Line 214: Consider moving RC27 to panel b instead of c to be in order of discussion.

Response: The cores from the literature are ordered from south to north. For the comparison with these cores we want it ordered regarding their location in the Arabian Sea. However, as the core site in our study is located in the north, we will change the order as follows: panel a) SL167 (Oman Margin, Gulf of Oman, this study), b) MD04-2876 (northern Arabian Sea), c) RC27-14 (Oman Upwelling) and d) NIOP906 (Somali Upwelling). We will add the location names (northern Arabian Sea, Oman Upwelling etc.) according to the later comment on figure 6 below.

Line 244: start new paragraph

Response: We agree with the referee and will start a new paragraph.

Line 244: "Gulf of Oman d15N record"

Response: We will add "$\delta^{15}N$" to the sentence and refer to Figure 6b (previous Fig. 5b), where the $\delta^{15}N$ record is shown.

Line 246: "was not present"

Response: We will rewrite the sentence to "This is in contrast to other AS records, not showing a prominent triple peak from D/O events 3 to 5 but 5 to 7 with a broader D/O 8 event beforehand (Schulz et al., 1998).".

Line 322: What does LTG stand for? latitudinal insolation gradient? in the line below it is abbreviated as LIG

Response: We are sorry for the confusion. This was a typo. It should be "latitudinal temperature gradient (LTG, obliquity signal" instead of "latitudinal insolation gradient". We will corrected this.

Figures:

Fig. 1: a) source for directions of wind pattern? satellite?

b) is this an annual average or seasonal data?

Response: a) We will add the reference "after Clift & Plumb, 2008; Hunt et al., 2018).

b) We will add "…at the end of the SW-monsoon season in September 2007…" to the description of panel b to highlight, that the shown CTD data set is a snapshot obtained during the M74/1b cruise.

Fig. 3: 14C (superscript 14)

Response: We will change this.

Fig. 4: lycopane instead of lycopene

Response: We will change the typo.

Fig. 5: brown line in (d) appears to be grey.

why is arrow with OMZ not extended to panel a?

Response: We will change the color of the line from brown to red for a better visibility. The OMZ arrow is not extended to panel a for several reasons. First, the diversity of benthic foraminifera depends on the bottom water oxygen content but also on the availability of food (see Schmiedl et al. 2023 and references therein). The food flux plays an important role, low food flux increases the competition on the sea floor and excludes the infauna leading to low diversity. Highest food flux also lead to low diversity because increasing physiological stress due to oxygen consumption and/or the dominance of few species grazing the food (see Levin et al., 2001). Because of that there is no linear link between the bottom water oxygenation and the diversity. Second the OMZ arrow is referred to the oxygen condition in the water column. We will add the latter to the figure description to make it clear. Further we will add another figure to show the relationship of the diversity of benthic foraminifera with oxygen and food supply.

Fig. 6: what does "mostly based on …" mean?

space missing after Fleitmann et al. (2009).

Response: We will add the space after the reference. We used "mostly based on" because there is a gap in the Sofular Cave record during the occurrence of D/O 2 (Fleitmann et al., 2009) .

It would help the reader to either have a small map inserted in this figure with the core locations or add "northern Arabian Sea/ Somali Upwelling/ Oman Upwelling/ Gulf of Oman" to the core sites.

Response: We will add the locations to the core description.

Fig. 7&8: Can the authors describe the colours in panel b?

Response: The reddish colors showing higher power of the wavelet power spectrum, whereas lower power is represented by blueish color. We will add this description to the figures as follows "The colours in b) represent the power of the wavelet power spectrum, with red (blue) showing high (low) power.".

Table:

Table 1: superscript 14 in 14C

Response: We will change this.

References mentioned in the reply:

[revised manuscript text omitted]

North Greenland Ice Core Project members: High-resolution record of Northern Hemisphere climate extending into the last interglacial period, Nature, 431(7005), 147–151, doi:10.1038/nature02805, 2004.

Pancost, R. D. and Boot, C. S.: The palaeoclimatic utility of terrestrial biomarkers in marine sediments, Mar. Chem., 92(1), 239–261, doi:https://doi.org/10.1016/j.marchem.2004.06.029, 2004.

Reichart, G. J., den Dulk, M., Visser, H. J., van der Weijden, C. H. and Zachariasse, W. J.: A 225 kyr record of dust supply, paleoproductivity and the oxygen minimum zone from the Murray Ridge (northern Arabian Sea), Palaeogeogr. Palaeoclimatol. Palaeoecol., 134(1), 149–169, doi:http://dx.doi.org/10.1016/S0031-0182(97)00071-0, 1997.

Rixen, T., Baum, A., Gaye, B. and Nagel, B.: Seasonal and interannual variations in the nitrogen cycle in the Arabian Sea, Biogeosciences, 11(20), 5733–5747, doi:10.5194/bg-11-5733-2014, 2014.

Rixen, T., Cowie, G., Gaye, B., Goes, J., do Rosário Gomes, H., Hood, R. R., Lachkar, Z., Schmidt, H., Segschneider, J. and Singh, A.: Reviews and syntheses: Present, past, and future of the oxygen minimum zone in the northern Indian Ocean, Biogeosciences, 17(23), 6051–6080, doi:10.5194/bg-17-6051-2020, 2020.

Rommerskirchen, F., Plader, A., Eglinton, G., Chikaraishi, Y. and Rullkötter, J.: Chemotaxonomic significance of distribution and stable carbon isotopic composition of long-chain alkanes and alkan-1-ols in C4 grass waxes, Org. Geochem., 37(10), 1303–1332, doi:10.1016/j.orggeochem.2005.12.013, 2006.

Schmiedl, G., Milker, Y. and Mackensen, A.: Climate forcing of regional deep-sea biodiversity documented by benthic foraminifera, Earth-Science Rev., 244(May), 104540, doi:10.1016/j.earscirev.2023.104540, 2023.

Shimmield, G. B., Price, N. B. and Pedersen, T. F.: The influence of hydrography, bathymetry and productivity on sediment type and composition of the Oman Margin and in the Northwest Arabian Sea, Geol. Soc. London, Spec. Publ., 49(1), 759–769, doi:10.1144/GSL.SP.1992.049.01.46, 1990.

Sigman, D. M., Altabet, M. A., McCorkle, D. C., Francois, R. and Fischer, G.: The δ15N of nitrate in the Southern Ocean: Nitrogen cycling and circulation in the ocean interior, J. Geophys. Res. Ocean., 105(C8), 19599–19614, doi:https://doi.org/10.1029/2000JC000265, 2000.

Szuman, M., Berndt, C., Jacobs, C. and Best, A.: Seabed characterization through a range of high-resolution acoustic systems – a case study offshore Oman, Mar. Geophys. Res., 27(3), 167–180, doi:10.1007/s11001-005-5999-0, 2006.

Tesdal, J.-E., Galbraith, E. D. and Kienast, M.: Nitrogen isotopes in bulk marine sediment: linking seafloor observations with subseafloor records, Biogeosciences, 10(1), 101–118, doi:10.5194/bg-10-101-2013, 2013.

Uchupi, E., Swift, S. A. and Ross, D. A.: Morphology and Late Quaternary sedimentation in the Gulf of Oman Basin, Mar. Geophys. Res., 23(2), 185–208, doi:10.1023/A:1022408106382, 2002.

UNEP: Overview on Land-Based Sources and Activities Affecting the Marine Environment in the ROPME Sea Area - UNEP Regional Seas Reports and Studies No. 168, [online] Available from: https://wedocs.unep.org/20.500.11822/31454, 1999.

Vogts, A., Moossen, H., Rommerskirchen, F. and Rullkötter, J.: Distribution patterns and stable carbon isotopic composition of alkanes and alkan-1-ols from plant waxes of African rain forest and savanna C3 species, Org. Geochem., 40(10), 1037–1054, doi:10.1016/j.orggeochem.2009.07.011, 2009.

---

## Author Comment (AC2)

**Comments Anonymous Referee #2**

The paper by Burdanowitz et al. aims to reconstruct oxygen conditions in the Gulf of Oman over the last 45,000 years using a multiproxy approach (bulk sediment δ15N, lipid biomarker analysis, and a benthic foraminifera index). The authors propose two distinct modes of oxygenation during this time, with the Holocene reflecting stable conditions, and other periods (e.g., MIS 3) displaying a high degree of fluctuations between oxygenated and deoxygenated conditions. The research contributes to our understanding of the connections between bottom water oxygenation in the region and global climate events.

The authors employ established methodology to produce a robust, high resolution record, and they present their findings in a cohesive manner. Overall, the manuscript is well-written, engaging, and well-organized, and is within the scope of the journal. I present some minor suggestions where additions may help clarify some ideas.

**Major points:**

Point 1: the δ15N sedimentary record is good, but the caveats of the method should at least be mentioned. Robinson et al. (2012) addresses when it is appropriate to use this method, versus when degradation of material becomes and important factor. The shallow water depth of the core used in this study is one factor that argues against alteration of the signal, but a time period of low accumulation could account for some changes in δ15N and should be addressed. Other useful references are Tesdal et al., (2012) and Junium et al., (2015).

Response: First, we thank the referee for her/his comments and suggestions, which helps us to improve the manuscript.

We agree that we initially did not address the limitations using sedimentary $\delta^{15}N$ in a proper way. We therefore will add following paragraph to the introduction: "Denitrification occurs in oxygen-depleted oceanic zones, serving as oxygen source for heterotrophic bacteria reducing nitrate via nitrite to $N_2$ (Devol, 2008; Rixen et al., 2020 and references therein). For these heterotrophic bacteria it is energetically more efficient to use the lighter $^{14}N$ from the nitrate source (Devol, 2008). Because of this isotopic fractionation, the remaining nitrate, which is used by phytoplankton, becomes more enriched in $^{15}N$ (Montoya, 2008 and references therein). The global average $\delta^{15}N$ of deep water nitrate is about 4.8 ± 0.2 ‰ (Sigman et al., 2000) but denitrification processes lead to more enriched $\delta^{15}N$ in nitrate with values above 18 ‰ in the Arabian Sea OMZ (Rixen et al., 2014). During the nitrate consumption, phytoplankton also uses the lighter $^{14}N$ and have an isotope discrimination factor of about 5 ‰ and incorporate the denitrification signal into the biomass (Montoya, 2008). Sinking of this biomass to the ocean floor transports the denitrification signal to the sediments (e.g. Altabet et al., 1995; Gaye-Haake et al., 2005). Therefore, $\delta^{15}N$ values can be used as a proxy for the OMZ strength in the water column (e.g. Altabet et al., 1995; Reichart et al., 1997). However, diagenetic processes can alter sedimentary δ15N values especially under low sedimentation rates (Gaye-Haake et al., 2005; Jung et al., 1997; Junium et al., 2015; Möbius et al., 2011; Tesdal et al., 2013). Nevertheless, it was found that in the Arabian Sea, especially under high sedimentation rates, sedimentary $\delta^{15}N$ values are a reliable indicator for past denitrification processes (Möbius et al., 2011)."

Sedimentation rates in SL167 are high, ranging between 9.7 and 49 cm/kyr. From other studies (e.g. Möbius et al., 2011 and Jung et al., 1997) we know, that with this high sedimentation rates it is

unlikely that $\delta^{15}N$ is altered by diagenesis. We will add this to the discussion. Further, we will add cross plot of TN- $\delta^{15}N$ and sedimentation rates – $\delta^{15}N$ to the appendix.

Point 2: the use of acronyms in the manuscript was confusing. Some acronyms were introduced (e.g., EBFOI) only for them to be written in full several paragraphs later. Other acronyms were only introduced in the last paragraph of the discussion (LTG and LIG, lines 322 and 323) and were not necessary. Some acronyms were not written out in full at all (ISM, line 265; AS, line 245). My suggestion would be to limit acronym use to terms that are frequently used throughout the paper, and to write out the full names of indices/events that are referred to only once or twice.

Response: We agree that some of the acronyms can be avoided due to their limited usage and we will check this. Further, we realized that we made a mistake regarding LTG and LIG. LTG is the latitudinal temperature gradient and not "latitudinal insolation gradient" as written in the manuscript and we will correct this.

**Minor points and typos:**

Line 9: "Climatic conditions and its change" -> "Climatic conditions can change"

Response: We will rewrite this sentence to "Changing climatic conditions can shape the strength and extent of the oxygen minimum zone (OMZ).".

Line 10: "for their ecosystem" -> "for its ecosystem"

Response: We will change it to "for its ecosystem".

Line 17: "Contrary -> "In contrast"

Response: We will change it to "In contrast".

Line 28: What is the largest sink? OMZs are the largest sink for nitrogen?

Response: The denitrification is the largest sink for nitrogen in the ocean. We will clarify this by adding following part to the sentence: "…and denitrification also acts as the largest sink...".

Line 29: N2O not N20

Response: We will change the typo.

Line 59: Why is the isoprenoid hydrocarbon lycopene rarely used? Any reasons you can point to?

Response: We will add "…which is mainly limited to sediments from OMZs or oceanic anoxic events in the past.." as well as the references Farrington et al. (1988), Dummann et al. (2021) and Sabino et al. (2021) to this sentence. A further explanation for its rarely usage is the fast degradation under oxic conditions as stated in lines 67-69 of the manuscript. Lycopane is also not found in every sediment core within the OMZ. For instance, in a core from the NE Arabian Sea (SO90-63KA) within the OMZ we could not find lycopane in the samples, but a "normal" terrestrial plant-wax derived *n*-alkane distribution (see Burdanowitz et al. 2021).

Section 2: RSW is also important in the southern Indian Ocean and has been detected in the Agulhas Current region (see Marshall et al., 2023); not necessarily relevant to your paper, but if you want to emphasize the importance of RSW, a sentence or two could be added here.

Response: The referee is correct, that RSW is also quite important for the southern Indian Ocean and Agulhas region. However, as we focus on the northern Arabian Sea we think this is beyond the scope of our manuscript.

Line 111: "The age-depth model (Figure 3)"

Response: Figure 3 (figure 4 in the new version of the manuscript) shows the results of the age-depth model. As we are describing the methods in this part, we will refer to the figure in the results section.

Section 3.2 and 3.3: Any references used for δ15N and column chromatography methods would be good here.

Response: We will add a reference (Menzel et al., 2014) describing the procedure for the $\delta^{15}N$ measurements. For the column chromatography of the lipids, we will add the reference Herrmann et al. (2016) and some further detailed description of the procedure as follows: "Therefore, a combusted Pasteur pipette was packed with cleaned cotton wool and about 2 cm $NaSO_4$. The column was then first cleaned with about 8 ml hexane. Then TLE was transferred with about 1 ml hexane to the column and the remaining 4 ml vial was cleaned 3 times with about 1 ml hexane, which was also transferred to the column. For the hexane-insoluble fraction, about 1 ml of DCM was added for 4 times on the column. The hexane-soluble was saponified (85°C, 2 h) in a 5 % potassium hydroxide (KOH) in MeOH solution and the neutral fraction was extracted with hexane.".

Line 130: "were" -> "was"

Response: We will change "were" to "was".

Line 150: "were counted at each depth from representative splits"

Response: We will add "at each depth" to the sentence.

Line 155: Could you explain the Shannon Index here briefly in one sentence?

Response: We will add "For benthic foraminiferal diversity, the Shannon Index H(S) was calculated according to Buzas and Gibson (1969). The H(S) considers the number of species and their relative proportion in the sample. The H(S) value is at a maximum, when all species have equal proportions, while species with low abundances contribute little to it. In eutrophic to mesotrophic ecosystems, the diversity and microhabitat structure is oxygen-controlled, as predicted by the TROX (Trophic-Oxygen) model (Gooday, 2003; Jorissen et al., 1995) (Figure 3)." for a brief explanation.

Line 157: A table showing the species and their habitat preference (oxic, suboxic, dysoxic) would be very helpful, even if for the supplemental.

Response: We will add a supplementary table showing the species names, preference to certain oxygen conditions (oxic, suboxic, dysoxic), and relative abundances in all studied samples.

Line 130: "were" -> "was"

Response: We will change this.

Line 233: "led" -> "lead"

Response: We will change this.

Line 235: This could use a reference: "the sea level was in general lower during MIS 3 than during the Holocene."

Response: We will add the references Rohling et al. (2008) and Siddall et al. (2003). As we refer to the Red Sea sea level curve shown in Figure 6f the sentence before, we will add the references only.

Line 240: "In total, the interplay and the bipolar seesaw structure of the northern and southern hemispheric climate signals may lead to the feature that some of the D/O events as well as Heinrich events are not represented in the record." This isn't very clear to me - do you mean the interplay between northern and southern signals? What specific combination of climate factors could result in the dampened signal in the region?

Response: The referee raises here an important question. It is not easy to answer it because local, regional and global factors have an impact on oxygenation in the water column and bottom water. We have tried so summarize these points in this section (lines 228 – 239). First, D/O events are different in their durations and strength (e.g. Buizert & Schmittmer, 2015) and thus their impact on the Indian and Asian Monsoon systems. Second, fluctuations of Red Sea sea level results in variations of RSW influence at the core site. Third, the northward extension of the AAIW is linked North Atlantic climate. In line 240 we mean the interplay of all these mentioned points, this includes the bipolar seesaw pattern between northern and southern hemispheric climate but also the climate and oceanic patterns on regional scale. We will rewrite this sentence for better understanding.

Line 245: You could reference your figure after mentioning the triple peak for the first time, as well as writing which proxies show the triple peak most clearly.

Response: We will add "$\delta^{15}N$" to the sentence and refer to Figure 5b, where the $\delta^{15}N$ record is shown.

Lines 228 – 277: This paragraph is quite long and puts forward several ideas. Maybe it could be broken up into two or three paragraphs (e.g., break at line 246 or 263).

Response: We agree with the referee and we will restructure this section. We will split this part into two main sections. The first one for the time period Pleistocene until Younger Dryas, with own paragraphs for the LGM and B/A and the other main section for the Holocene.

Line 276: To what does "this" refer to? The fact that no lycopanes were preserved? Why does the presence of this species aid the interpretation (because of its habitat preference)?

Response: In general, high ratios of (lycopane + $C_{35}$)/$C_{31}$ indicate a good preservation of lycopane under oxygen depleted conditions. In this case also fast burial of organic matter, indicated by high total organic carbon mass accumulations rates (Fig A1a), plays an important role of the high lycopane content. "This" refers to the assumption that high total organic carbon mass accumulation rates are responsible for the good preservation of lycopane. Further, the dominant *species Uvigerina peregrina* favors high supply/quantity of organic matter (Koho et al., 2008; Schmiedl et al., 2010). We will add this information.

Line 289: "environmental" -> "environmentally"

Response: We will change this.

Line 299: Periods just before this were written to one decimal point, and the ones here are written with two; it might be clearer to round these to one point.

Response: We agree with the referee and will round it to one decimal point.

Line 308: What does an interstadial AMOC mode mean in this context? A weaker circulation?

Response: Buizert & Schmittner (2015) use the term "interstadial AMOC mode" to a strong AMOC/warm North Atlantic Ocean. In contrast, a "stadial AMOC mode" would mean a weak AMOC/cold North Atlantic. They also describe a "glacial AMOC mode" associated with the cold Heinrich events and weakest AMOC as well as strongly reduced Glacial North Atlantic Intermediate Water formation. We will add this information to make it clear as follows "…inhibited an interstadial AMOC mode (strong AMOC/warm North Atlantic), which is required to initiate D/O events (Buizert and Schmittner, 2015).".

Line 310: This phrase sounds odd, as it is not explored further. Perhaps a better way to phrase it would be "this pattern may or may not be the reason for the absence of a strong D/O 6 signal…"

Response: We used "might not be" in the wrong context. We will change it.

Line 334: the phrasing could be changed - "the very well ventilated conditions" -> "the well-mixed/the strong ventilation/the oxygenated conditions"

Response: We will change it to "The well oxygenated conditions…".

Line 344: the phrasing could be changed – "with swinging back and forth oxygen conditions" -> "with fluctuating high and low oxygen conditions."

Response: We will change it as suggested.

References mentioned in the reply:

[revised manuscript text omitted]